# Estimation of the state and parameters in ice sheet model using an ensemble Kalman filter and Observing System Simulation Experiments

Youngmin Choi<sup>1</sup>, Alek Petty<sup>1</sup>, Denis Felikson<sup>2</sup>, and Jonathan Poterjoy<sup>3</sup>

<sup>1</sup>Earth System Science Interdisciplinary Center, University of Maryland, College Park, MD, USA

<sup>2</sup>NASA Goddard Space Flight Center, Greenbelt, MD, USA

<sup>3</sup>Department of Atmospheric & Oceanic Science, University of Maryland, College Park, MD, USA

Correspondence: Youngmin Choi (yochoi@umd.edu)

**Abstract.** Better constraining the current and future evolution of Earth's ice sheets using physical process models is essential for improving our understanding of future sea level rise. Data assimilation is a method that combines models with observations to improve current estimates of model states and parameters, leveraging the information and uncertainties inherent in both models and observations. In this study, we present an ensemble Kalman filter-based data assimilation (DA) framework for ice sheet modeling, aiming to better constrain the model state and key parameters from a single semi-idealized glacier domain. Through a synthetic twin experiment, we show that the ensemble DA method effectively recovers basal conditions and the model state after a few assimilation cycles. Assimilating more observations improves the accuracy of these estimates, thereby improving the model's projection capabilities. We also utilize Observing System Simulation Experiments (OSSEs) to explore the capabilities of the ensemble DA framework to assimilate different types of data and to quantify their impact on the model state and parameter estimation. In our experiments, we assimilate land ice elevation data simulated based on The Ice, Cloud, and Land Elevation Satellite-2 (ICESat-2) products. These experiments are crucial for identifying observations with the largest impact on state and parameter estimates. Our assimilation results are highly sensitive to design choices for observation networks, such as spatial resolutions and prescribed uncertainties. Notably, the marginal improvements or increases in RMSE observed at coarser resolutions suggest that, beyond a certain spatial threshold, additional observations do not improve and may even degrade long-term estimates of model parameters and state. The ensemble DA framework, capable of assimilating multi-temporal observations, shows promising results for real glacier applications through a continental ice sheet model. Additionally, this framework provides a flexible infrastructure for performing OSSEs aimed at testing various observational settings for future missions, as it requires less numerical model re-development than variational methods.

### 1 Introduction

The combined contribution of the Greenland Ice Sheet (GrIS) and the Antarctic Ice Sheet (AIS) to global sea level is one of the most significant sources of uncertainty in projections of sea-level rise for the coming century (Intergovernmental Panel on Climate Change (IPCC), 2023). In recent years, numerical ice sheet models have significantly advanced through improved

ice flow physics, enhanced spatial resolution, and their ability to simulate moving boundaries (Nowicki and Seroussi, 2018). Despite these advancements, projections of mass change for both the AIS and GrIS, and consequently their contributions to sea-level rise over the coming century, exhibit significant spread, primarily due to uncertainty in key model parameters and model initialization (Nowicki et al., 2016; Seroussi et al., 2020; Goelzer et al., 2020).

Data assimilation (DA) methods for ice sheet modeling generally fall into two categories: snapshot and transient inversions (Choi et al., 2023), which use single-time observations and time series of observation, respectively. Snapshot inversion, implemented using a form of variational data assimilation methods, has been widely used in ice sheet models to constrain basal conditions, such as the friction coefficient, and estimate the present state of the ice sheet using surface observations, such as surface velocity (MacAyeal, 1992; Morlighem et al., 2010). However, these approaches generally rely on observations at a single time to perform time-independent inversions of model parameters. This method captures a specific state of the ice sheet at a particular time (Morlighem et al., 2013; Gillet-Chaulet et al., 2012), but it often introduces nonphysical artifacts into the model's initial state, potentially propagating artifacts into transient simulations rather than capturing actual trends of changes in ice dynamics (Seroussi et al., 2011; Goldberg et al., 2015). Such artifacts in initial conditions could affect model simulations over centuries to millennia due to the slow response time of ice sheets (Seroussi et al., 2019).

Alternatively, data assimilation techniques that leverage time-varying surface observations have been developed to more accurately constrain ice flow over longer periods. The use of computational techniques such as automatic differentiation in ice sheet models (Goldberg and Heimbach, 2013) has enabled the assimilation of more observations into transient model simulations, thereby capturing the model evolution during the assimilation period—the time window during which observations are assimilated into the model. While this method has been applied in regional modeling studies (Larour et al., 2014; Goldberg et al., 2015; Choi et al., 2023), scaling time-varying data assimilation approaches for simulations covering entire ice sheets remains challenging due to the complexities involved in developing a time-dependent adjoint model and the substantial memory requirements of automatic differentiation (Choi et al., 2023). Furthermore, this method as well as static inversion do not explicitly compute the uncertainty coming from the model state and parameters (Carrassi et al., 2018).

Ensemble data assimilation methods, which use an ensemble of model simulations, offer an alternative to variational approaches by explicitly representing uncertainty in the model state and parameters while assimilating time-varying observations. Various forms of ensemble Kalman filter (EnKFs) have been effective for assimilating diverse observations into complex, large-scale and non-linear geophysical models (Carrassi et al., 2018). The underlying principle of the Kalman filter involves the sequential assimilation of data to estimate state variables for numerical models. This is achieved by iteratively adjusting the model state to better represent the unknown 'true' state of the system based on noisy observations (Carrassi et al., 2018). The assimilation based on the EnKF is carried out across an ensemble of model runs, each representing plausible system states. As new observations are incorporated within the assimilation period, the ensemble mean presents an increasingly more accurate estimate of the model state. When the model state is updated at each assimilation time, the model parameters can also be updated alongside state variables to reflect past and present observations (Iglesias et al., 2013). Unlike time-independent inversions relying on a snapshot of observations from a single time, this framework enables the use of a time series of observations to provide an improved estimate of the model state and parameters. While transient inversions also assimilate time-varying

observations, they typically estimate a single model state conditioned on all observations within the assimilation window. In contrast, the EnKF updates the model state sequentially at discrete observations time, without the need for estimating a tangent linear and adjoint for the model and measurement operators. In addition, EnKFs, similar to the classic Kalman filter, provide a direct estimate of uncertainty in model state and parameter estimates, which is represented heuristically through the sample error covariance.

The ice sheet modeling community has traditionally relied on snapshot inversion methods based on adjoint-based techniques for parameter estimation, using time-invariant mosaics or composite data (e.g., multi-year averaged surface velocity fields; Morlighem et al. 2010). Compared to these methods, ensemble DA approaches have been less commonly used in ice sheet modeling, primarily due to historical limitations in observational data, computational cost, and the challenges of representing uncertainty in ice sheet models. Ensemble approaches rely on time-varying observations with well-characterized uncertainties, but surface observations for ice sheets have often lacked reliable uncertainty estimates, making them less suitable for ensemble DA. Additionally, ensemble methods typically require multiple forward model runs, making them more computationally demanding than snapshot inversion approaches. Another limitation is that poorly understood or unquantified errors in the ice flow model itself may limit the reliable estimation of covariances using ensemble statics.

Nonetheless, promising results have been demonstrated in recent studies that apply ensemble DA to estimate both the model state and basal conditions of ice sheets (Bonan et al., 2014, 2017; Gillet-Chaulet, 2020). These include initializing marine ice sheet models that incorporate ice fronts and grounding line migration. However, these studies (Bonan et al., 2014, 2017; Gillet-Chaulet, 2020) utilized simplified flowline models, limiting the representation of the horizontal stress field that can impact ice dynamic processes through, for example, buttressing. Such unrepresented physics in ice flow models or structural model errors may limit the reliable estimation of covariances. As more complete, time-resolved datasets with robust uncertainty estimates become available, and as ice sheet models grow more sophisticated while computational costs continue to decrease, ensemble DA methods are increasingly worth exploring for larger-scale, more realistic ice sheet models.



Data assimilation and associated data denial experiments—where the impact of specific observations is evaluated by temporarily removing them from the assimilation process—can be used to test the benefit of current observations, typically referred to as Observation System Experiments (OSEs), as well as to evaluate the potential benefits of proposed observations, typically referred to as Observation System Simulation Experiments (OSSEs, Arnold Jr and Dey, 1986; Masutani et al., 2010). The main difference is that OSEs assimilate real observations, while OSSEs assimilate synthetic observations generated from model output with errors sampled from a prescribed observation error distribution representative of real measurement uncertainties. Both approaches aim to provide a systematic assessment of the value of observations for improving model state and parameter estimation. OSSEs have been successfully applied to atmospheric and oceanic models for decades, where analysis systems and the required DA frameworks are far more established (Boukabara et al., 2016; Hoffman and Atlas, 2016). For ice sheet modeling, however, the application of these OSE/OSSE approaches has, to our knowledge, not been previously explored.

This study explores the feasibility and benefits of using an EnKF to assimilate surface observations into a 2D plan-view ice model, with the aim of accurately estimating both the model state (ice thickness) and key model parameters related to basal conditions (basal friction and topography). Using the shelfy-stream approximation (SSA, MacAyeal, 1989) for the stress

Figure 1. Schematic overview of the ensemble data assimilation workflow using the EAKF within the DART-ISSM framework.

balance of the ice sheet, ice thickness serves as the only prognostic variable representing the model state. Basal friction and topography, which cannot be directly measured, are treated as key model parameters that must instead be estimated through surface observations. We perform a twin experiment in which we evaluate the estimated model state and parameters by comparing them with true values and using them as initial conditions to assess the impact of ensemble data assimilation on model projections (Section 2.3). Our modeling settings are similar to those used in the previous study (Gillet-Chaulet, 2020), which used a flowline model, with necessary modifications for our model domain geometry and the coupling between a 2D ice sheet model and the data assimilation system (Section 2.1). We investigate various ensemble DA parameters on a synthetic ice sheet domain to explore effective ensemble DA strategies relevant to ice sheet modeling (Section 2.2). One of the primary objectives of this research is to use an idealized model configuration to help inform future efforts in applying an EnKF for real glacier cases. Within this context, we also configure OSSEs to evaluate the impact of various configurations of observations on the estimated model state and basal conditions (Section 2.4).

### 2 Methods


This section describes the ice sheet model configuration (Section 2.1), the ensemble DA framework (Section 2.2), and the experimental designs used in this study (Section 2.3 and 2.4). We first outline the twin experiment setup, which tests the ability of the DA framework to recover the model state and parameters under idealized conditions. We then describe the OSSEs, which explore the effects of different observational strategies on model initialization. Our methods are summarized in Figure 1.

### 2.1 Model Setup

We use the Ice-sheet and Sea-level System Model (ISSM, Larour et al., 2012) to simulate the model state and forecast its evolution over time. ISSM is a parallelized finite element ice flow model with anisotropic mesh refinement capabilities, which allows efficient ensemble simulations of ice sheets.

Table 1. Parameters for the reference ice sheet domain


| Parameter    | Value  | Description                                           |
|--------------|--------|-------------------------------------------------------|
| $z_{b,deep}$ | -720 m | Depth of the bedrock topography                       |
| $L_x$        | 640 km | Domain length (along ice flow)                        |
| $L_y$        | 80 km  | Domain width (across ice flow)                        |
| $d_c$        | 500 m  | Depth of the trough compared with the side walls      |
| $w_c$        | 24 km  | Half-width of the trough                              |
| $f_c$        | 4 km   | Characteristic width of the side walls of the channel |

We construct our reference simulation using a bed geometry inspired by Asay-Davis et al. (2016) and Gillet-Chaulet (2020). The synthetic bed topography features large-scale overdeepening combined with added small-scale roughness. The general shape of the bed is defined as:

$$z_b(x,y) = \max [B_x(x) + B_y(y), z_{b,deep}]$$
 (1)

$$B_x(x) = \begin{cases} 150 - 3x, & 0 \text{ km} \le x \le 350 \text{ km} \\ -900 + 5(x - 350), & 350 \text{ km} \le x \le 450 \text{ km} \\ -400 - 3(x - 450), & 450 \text{ km} \le x \le L_x \end{cases}$$
 (2)

$$B_y(y) = \frac{d_c}{1 + e^{-2(y - L_y/2 - w_c)/f_c}} + \frac{d_c}{1 + e^{2(y - L_y/2 + w_c)/f_c}}$$
(3)

where the parameter values used in these equations are given in Table 1. Following Gillet-Chaulet (2020), we introduce small-scale roughness to the bed topography using a midpoint displacement method (Fournier et al., 1982). This method generates a two-dimensional surface by iteratively subdividing a grid, assigning random heights to the corners, and displacing midpoints with added random displacement. The magnitude of the displacement is scaled by a standard deviation that decreases with each iteration as  $2^{0.5H}$ , where H is the roughness factor, set to 0.7 in this study. We apply this method at 100 m resolution with 10 recursive subdivisions, starting from an initial standard deviation of 500 m. This process produces an asymmetrical bed topography, which may better reflect realistic subglacial features, although we conduct an idealized twin experiment in this study. The model domain spans 0 to 640 km in the x-direction and 0 to 80 km in the y-direction. This domain is discretized into approximately 27,000 elements using a triangular mesh with resolutions varying from 500 m near the coast to 10 km inland.

The basal friction coefficient follows a sinusoidal function similar to that used by Gillet-Chaulet (2020), comparable to the inferred friction coefficient in Thwaites Glacier of Antarctica in terms of both amplitude and spatial variations (Brondex et al., 2019; Gillet-Chaulet, 2020). In this study, we have adjusted this function for a 2D domain with an additional y-component (Fig. 6a):

$$C(x,y) = C_x(x) \times C_y(y) \tag{4}$$

$$C_x(x) = 0.02 + 0.01 \sin\left(5 \frac{2\pi(x - L_x)}{L_x}\right) \sin\left(30 \frac{2\pi x}{L_x}\right)$$
 (5)

$$C_y(y) = \sin\left(\pi \, \frac{(y - L_y)}{L_y}\right) \, + \, 2 \tag{6}$$

where  $C_x$  and  $C_y$  are the x and y components of a friction coefficient (C), respectively.

For the stress balance of an ice sheet, we use the shelfy-stream approximation (SSA, MacAyeal, 1989), which simplifies the Stokes equations for cases with a small aspect ratio and basal friction. The basal stress,  $\tau_b$ , is described by the Weertman friction law for grounded ice:

$$\tau_b = C|u_b|^{\frac{1}{m}-1}u_b \tag{7}$$

where  $u_b$  the ice basal velocity, and m the velocity exponent set to 1/3 in this study.

The ice viscosity is defined using Glen's law (Glen, 1955):



$$\mu = \frac{B}{2\dot{\varepsilon}_e^{1-\frac{1}{n}}},\tag{8}$$

where B is the ice viscosity parameter,  $\dot{\varepsilon}_e$  the effective strain rate, and n Glen's law exponent set to 3.

The position of the ice front is fixed at the end of the domain, and the evolution of the grounding line is simulated with a subelement grounding line parameterization (Seroussi et al., 2014).

We run the model until it reaches a steady state using a uniform surface accumulation rate of 0.3 m/yr, without any basal melting. After 25,000 years, the ice sheet stabilizes at a steady state, with a grounding line located approximately at x = 470 km along the center line of the glacier, just downstream of the region of overdeepening (Fig. 2). To introduce dynamic changes, we perturb this equilibrium state by instantaneously reducing the surface mass balance to -0.3 m/yr. We also introduce basal melting using a simple melt-depth parameterization, as described by Favier et al. (2014), setting the melt rate of 200 m/yr at a depth of 800 m, which results in an actual melt rate of approximately 170 m/yr beneath the ice shelf. Although this melt rate exceeds observed present-day basal melt rates, we choose this value to create a strong dynamical response over a forecast period, ensuring that the effects of data assimilation could be clearly evaluated. The elevated melt rate is not intended to represent a realistic present-day scenario, but rather to serve as a diagnostic tool in the context of a twin experiment described in Section 2.3. To generate the reference simulation for our experiments, we run the model for an additional 200 years, while keeping these surface and basal forcings constant. For the first 100 years, the grounding line retreats at a relatively slow pace, but the retreats accelerate after approximately 130 years (Fig. 2b). We refer to this simulation as the "reference simulation", from which we derive synthetic observations and reproduce the state and parameters through our ensemble DA framework. The setup of the reference simulation resembles an idealized Antarctic glacier.

**Figure 2.** (a) Initial steady-state ice surface elevation. (b) Bed topography of model domain and grounding line positions every 10 years from 0 to 200 years for the reference simulation (white to red). (c) Initial steady-state velocity with contours of 50 m/yr and 100 m/yr (magenta lines). The white line shows the initial grounding line position. (d) Mesh resolution of the model domain.

### 2.2 Data assimilation



We use the Data Assimilation Research Testbed (DART, Anderson et al., 2009) to implement ensemble data assimilation with ISSM. DART provides various DA algorithms and modules to create a complete end-to-end DA framework. In this study, we utilize the Ensemble Adjustment Kalman Filter (EAKF, Anderson, 2001) algorithm within DART, which belongs to a class of deterministic ensemble square-root filters (Tippett et al., 2003). In contrast to the standard stochastic EnKF—which perturbs observation-space quantities randomly for each ensemble member to account for observational uncertainty—the EAKF avoids additional perturbations and instead analytically adjusts the ensemble members to match the posterior mean and covariance determined by the original Kalman filter equations (Anderson, 2001). This approach improves numerical stability and reduces sampling noise over stochastic EnKFs, especially for small ensemble sizes (Whitaker and Hamill, 2002). In this study, we choose the EAKF due to its reduced sensitivity to ensemble size and improved robustness in geophysical systems, as demonstrated in previous studies using DART (Zubrow et al., 2008; Anderson et al., 2009). Throughout this paper, we use "EnKF" to refer to ensemble Kalman filter methods more generally, and "EAKF" to refer specifically to the version implemented in this study. We refer readers to Anderson (2001) for the full algorithmic details of the EAKF.

Within DART, ice sheet variables are placed into a state vector, and the filter uses the ensemble-estimated error covariance to compute the Kalman gain needed to update the model state with available observations. The state vector is augmented to include both prognostic variables and model parameters to be estimated. Under the stress balance of SSA, the velocity is a diagnostic variable, and due to the flotation condition, ice thickness is the only prognostic variable (Gillet-Chaulet, 2020). In

this study, the state vector includes ice thickness (state variable), and basal friction coefficient and bed topography (model parameters), allowing joint estimation of the model state and parameter fields through the DA process (Fig. 1).

A common challenge with EnKFs, including the EAKF used in this study, is the issue of undersampling, which arises when the size of the ensemble is significantly smaller than the independently observed degrees of freedom for the model state. Sampling errors occur because the ensemble-based covariance is only an approximation of the true covariance, and small ensembles may not adequately capture variability across the full state space (Carrassi et al., 2018). In our experiments, we use ensemble sizes of 30, 50, and 100, while the number of observations can range from hundreds to thousands, depending on the observation configuration. Localization and inflation are common methods to mitigate undersampling issues and increase the stability of the EAKF (Carrassi et al., 2018; Morzfeld and Hodyss, 2023). Localization adjusts the spatial influence of observations, thereby preventing the distortion of estimates by distant observations. While previous studies (Gillet-Chaulet, 2020; Cook et al., 2023) have explored the effects of localization on the model state estimation using flowline models, its application to 2D plan-view models remains unexplored. Similarly, inflation, which addresses sampling errors by artificially increasing the forecast covariance matrix, has not been thoroughly studied for large-scale ice sheet modeling. To identify the most effective settings, we conduct sensitivity tests using a range of both localization radii (2 to 20 km) and inflation factors (1.00 to 1.20) within our ensemble data assimilation framework.

## 2.3 Twin experiment







We conduct a twin experiment to evaluate the performance of using an EAKF to assimilate surface observations into a 2D planview ice model. Using the ISSM-DART DA framework, we aim to estimate the ice sheet state together with model parameters. Here, we assume that the friction coefficient and the bed topography are the only two unknown parameters that need to be estimated, while all other parameters and forcings (e.g., ice rigidity, surface mass balance) are perfectly known and identical to those used in the reference simulation. We assimilate annual surface observations derived from the reference simulation over a 30-year span—approximately the satellite observational period for ice sheets—to assess the ability of the ensemble DA framework to recover the initial model state and basal conditions of the reference ice sheet.

We obtain synthetic surface observations of ice elevation and velocities from the reference simulation and assume that the surface elevation and velocities are observed at annual resolution (e.g., at the start of each year) at each ISSM mesh node. To simulate observation error, we add uncorrelated Gaussian noise with a standard deviation of 5 m for the surface elevation and 10 m/yr for the velocity as a simple uncertainty baseline. These standard deviation values are lower than the ones from Gillet-Chaulet (2020), but still within a plausible range according to recent studies (Dai and Howat, 2017; Mouginot et al., 2017). Dai and Howat (2017) report vertical elevation uncertainties below 5 m in well-constrained regions, and Mouginot et al. (2017) report horizontal velocity uncertainties ranging from 5 - 20 m/yr depending on the region. We choose values at the lower end of these ranges to isolate the performance of the DA framework under favorable conditions. We explore the sensitivity to larger uncertainties in our OSSEs presented in Section 2.4.

To generate initial ensembles, we adopt an approach similar to that described by Gillet-Chaulet (2020). For the friction coefficient, we create a random field, assuming a known mean value of  $2,500 Pa m^{-1/3} a^{1/3}$  across the domain and using a

prescribed covariance model for spatial dependency. We use a Gaussian function for the variogram with a range of 5 km and a sill of 90,000. These values for the range and sill were selected based on Gillet-Chaulet (2020), with adjustments made for the domain and friction law used in this study. For bed topography, we use an exponential function for the variogram with a range of 50 km, a sill of  $4,000 \, m^2$  and a nugget of  $200 \, m^2$ , also based on the same study (Gillet-Chaulet, 2020). Unlike the friction coefficient, which typically cannot be directly measured and often lacks prior knowledge, the bed topography can be measured using ice penetrating radar (e.g., Evans and Robin, 1966; Dowdeswell and Evans, 2004; Rodriguez-Morales et al., 2014). We assume that we have radar measurements of bed topography along tracks perpendicular to ice flow every 30 km. Using kriging with an exponential covariance model, we generate a conditional random field of the bed topography constrained by these observations. Initial ensembles for both parameters are created using the GSTools Python package (Müller et al., 2022). Additional initial ice sheet variables, such as initial thickness and velocity, are calculated through a stress balance solution using the initial ensemble of friction coefficient and bed topography. In our setup, the basal friction coefficient and bed topography are estimated jointly as part of the augmented state vector. While we do not prescribe a prior correlation between them, the EAKF uses ensemble-based cross-covariance between these parameters and background variables to update both fields during the assimilation process.

To date, no studies have determined optimal localization and inflation factors for large-scale 2D ice sheet models. Therefore, we conduct sensitivity tests to identify the best values for these parameters across various ensemble sizes. For this study, a Gaspari-Cohn fifth-order polynomial is used for horizontal direction localization to limit observation updates within a specific radius (Gaspari and Cohn, 1999). Localization is applied to reduce correlations between model states projected into observation space and the unobserved state variables, which does not explicitly damp covariances across co-located variables (Anderson, 2007). For inflation, we use the spatially uniform state space inflation (Anderson et al., 2009). We explore various combinations of inflation and localization values to find the optimal combination. Specifically, we vary the localization radius from 2 to 20 km in 2 km increments and adjust the inflation factors from 1.00 to 1.20 in 0.02 intervals. Initial experiments begin with an ensemble size of 30, based on findings from smaller-scale flowline model studies that demonstrate robust DA performance with relatively small ensembles. We then extend our experiments to larger ensembles, using 50 and 100 members, to examine the impact of ensemble size on DA performance in large-scale ice sheet modeling.

To evaluate the effectiveness of the ensemble DA framework in retrieving basal conditions and ice sheet state, we calculate the root-mean-square error (RMSE) between the analysis mean states and the designated true values for bed topography (RMSE\_B), friction coefficient (RMSE\_C), and ice thickness (RMSE\_H). After each analysis, RMSE values are computed at all nodes where basal conditions have been updated through assimilation. This calculation includes only those nodes where at least one node in the triangular mesh is grounded, as surface observations only respond to changes in the basal condition of grounded ice.

Based on the model state and parameters estimated from the DA simulation, we conduct deterministic and ensemble forecasts extending up to t = 200 yr to explore the impact of ensemble DA initialization on model projections. We use the ensemble mean to initialize the deterministic simulation and the full ensemble to initialize the ensemble forecast simulations, similar to Gillet-Chaulet (2020). We also utilize the estimated model state and parameters as initial conditions from various points in the DA

simulation different initial conditions, e.g., the analyzed states at t = 5 yr, t = 15 yr, and t = 30 yr, for forecast simulations to investigate the impact of different DA periods on model simulations.

### 2.4 Observing System Simulation Experiments (OSSEs)






We conduct OSSEs within our synthetic model domain to investigate the potential impact of varying observed quantities and their associated uncertainties. For our OSSEs, we assume a "perfect" model without any model error, following the perfect model OSSE framework (Zhang et al., 2018). While the twin experiment described in the previous section is more focused on testing the capabilities of the EAKF under ideal conditions, the suite of experiments in this section is designed to explore the feasibility of performing joint state-parameter estimation for the ice sheet model under realistic observational settings, which will provide valuable insight and guidance for future, more realistic OSSE efforts. In this study, we primarily explore the impact of different types of surface elevation observations and their uncertainties. We assimilate the synthetic elevation data in two different ways: i) along ground tracks, which mimics The Ice, Cloud, and Land Elevation Satellite-2 (ICESat-2) ATL11 product, ii) at regularly gridded locations, which mimics the ICESat-2 ATL15 product (Smith et al., 2023, 2024). We use the same velocity data as in the previous twin experiment, assuming that the velocity products provide almost full coverage of annual velocity both spatially and temporally, and we focus on the impact of surface elevation observations.

For the along-track data, we generate synthetic surface elevation observations along tracks that emulate the Reference Ground Track (RGT) used by ICESat-2 ATL11 product. The RGT is a virtual line that corresponds to the nadir track of the designed orbit (Smith et al., 2019). For our synthetic domain, surface elevation is assumed to be observed annually, while the actual temporal resolution of ATL11 data is 91 days. Synthetic observations are spaced every 60 m along each track, which is the spatial resolution of ATL11 ice height data (Smith et al., 2023). While the actual ATL11 product exhibits varying cross-track spacing depending on latitude, we test cross-track spacings from 5 to 15 km, which covers the range of cross-track spacing of the ICESat-2 RGTs in the polar regions (Fig. 3). To generate synthetic observations, we linearly interpolate model surface elevation at surrounding mesh nodes to the observation points along our tracks. We also explore the impact of the observational uncertainties on the DA performance by conducting experiments with different levels of uncertainty in surface elevation. These experiments aim to determine the permissible level of error for different surface elevation products to ensure reliable DA for our model domain. We introduce Gaussian noise to surface elevation at each mesh node, using standard deviation ranging from 5 to 20 m with 5 m increments, and propagate standard errors to points along the tracks.

For gridded elevation observations, we create synthetic datasets at 1 km, 10 km and 20 km resolutions, corresponding to the spatial resolution of ATL15 product. The ATL15 product is a spatially continuous gridded dataset of land ice height-change (Smith et al., 2024). We first interpolate surface elevation from the mesh used in the reference simulation onto a grid with 100 m resolution, then average these 100 m grids to create a 1 km grid cell, using equal weights for all 100 m grids. Surface elevation data at 10 km and 20 km resolutions are created similarly from 1 km grid data. In our OSSEs, we assume an annual observation frequency of surface elevation for consistency across experiments, including the twin experiments, although the actual temporal resolution of ATL15 data is 91 days. Similar to the track elevation data, Gaussian noise is introduced with standard deviations from 5 to 20 m at each mesh node, with propagated error onto the gridded data.

**Figure 3.** Elevation observations taken along synthetic ground tracks from a configuration of (a) 5 km cross-track spacing, (b) 10 km cross-track spacing, and (c) 15 km cross-track spacing, with data points posted every 60 m along the track.

# 3 Results






# 3.1 Twin experiments and projections

Our twin experiments show the feasibility of the EnKF DA approach for ice flow modeling. The experiments are conducted with a range of configurations. Fig. 4 shows the RMSE values for the bed topography, friction coefficient, and ice thickness after 30 years of DA. As the ensemble size increases, DA performance remains relatively robust—demonstrated by lower RMSEs—over a wider range of localization radii and inflation factors. We observe that the best DA results, indicated by the minimum RMSEs, are achieved with a localization radius of 4 km for the friction coefficient and 6 km for bed topography and ice thickness. When the localization radius is set below those optimal values (4 km for friction coefficient and 6 km for bed topography and ice thickness), a significant increase in RMSEs occurs, and any increase beyond those optimal values also results in gradual increases in RMSEs. The optimal inflation factors tend to decrease as the ensemble size increases because larger ensembles generally provide better approximations of the true error covariance, reducing the need for artificially inflating the covariance to compensate for sampling errors (Anderson et al., 2009; Carrassi et al., 2018). For our experiments, optimal inflation values range between 1.10 - 1.14 for the friction coefficient and 1.16 - 1.18 for bed topography and ice thickness, when using the optimal radius for each parameter. Additionally, with the optimal localization radius, we note an improvement in DA performance with increasing inflation up to a certain threshold, beyond which the performance significantly decreases.

To assess the impact of ensemble size, we compare the evolution of RMSEs as a function of assimilation time using the optimal localization and inflation factors identified above (Fig. 5). For the friction coefficient, RMSE decreases rapidly during the first three years and continues to decrease steadily until the end of the assimilation window. The RMSE values of bed topography and ice thickness show a relatively steady decrease across all tested ensemble sizes, without an initial rapid drop. In all experiments shown in Fig. 5, the small increase in RMSE is examined during the early period of assimilation; however, as the assimilation continues, the RMSE values decrease again until the end of the assimilation period. The simulations with

**Figure 4.** The root-mean-square error (RMSE) between the analysis mean and the reference at t = 30 years as a function of the inflation factor and the localization radius for different ensemble sizes. (a-c) friction coefficient, (d-f) bed elevation, and (g-i) ice thickness. The grey indicates experiments that diverge by t = 30 years. The black box in each panel represents the location of minimum RMSE.

larger ensemble sizes show an improvement in DA performance compared to an ensemble size of 30, but the benefits saturate as the ensemble size increases from 50 to 100. For the remaining experiments in this study, for illustrative purposes, we proceed with an ensemble size of 50, a localization radius of 4 km, and an inflation value of 1.12.

The reference friction coefficient and bed topography, along with the ensemble mean fields, before and after assimilation with the DA configuration selected above, are shown in Fig. 6 and Fig. 7. We also show how the difference between true ice thickness and the ensemble mean changes before and after assimilation in Fig. 8. As more observations are assimilated, the discrepancies from the reference fields decrease compared to the initial ensemble mean. The areas around the grounding line, where the signal-to-noise ratio of velocity is relatively high, exhibit the most significant improvements through ensemble DA. In these regions, the spatial variations of both the friction coefficient and bed topography fields are accurately captured by the ensemble DA process. At the end of the 30-year assimilation period, areas located far upstream (up to 350 km) from the



**Figure 5.** The evolution of mean analysis RMSE for (a) friction coefficient, (b) bed topography, and (c) ice thickness using three different ensemble sizes. Each plot uses the localization radius and inflation factor that produce the minimum RMSE at t = 30 yr (Fig. 4).

**Figure 6.** (a) Reference friction coefficient (i.e., truth), (b) the ensemble mean friction coefficient before assimilation, (c)-(e) the ensemble mean friction coefficient after (c) 5 years, (d) 15 years and (e) 30 years of assimilation. The localization radius is set to 4 km and the inflation factor is 1.12 with the ensemble size of 50. The red lines show the grounding line positions.

grounding line continue to show improvements, although not as significant as those observed near the grounding line. The pattern in the estimated ice thickness is very similar to that of bed topography. The artifacts observed in bed topography and ice thickness are the result of the conditional random fields generated using the kriging method, which can produce "bull's eye" patterns commonly observed between observation points. In our model setup, surface elevation is defined as the sum of ice thickness and bed topography (surface = thickness + bed). Therefore, as surface observations are assimilated, improvements in bed estimates are reflected in the estimated thickness field.

Figure 7. Same as Fig. 6 but for bed topography.



Figure 9 presents the changes in ice volume over time for the reference simulation, along with the forecast simulations based on the ice sheet state with and without data assimilation over periods of 5 to 30 years. Forecast simulations were conducted in two ways, one with the ensemble mean model state and parameters for the single deterministic simulations and the other with the full ensemble members for the ensemble forecast. Without data assimilation, the deterministic forecast—using the ensemble mean basal conditions (e.g., initial mean basal conditions)—tends to underestimate ice loss over the 200-year period. This simulation, however, captures the accelerated volume loss observed in the reference simulation beginning at t = 130 yr, when the grounding line enters the reverse-sloping bed topography. By the end of the forecast simulation, the discrepancy in volume loss between the reference and deterministic simulations is 2,700 Gt. Across the ensemble members, the changes in ice volume at t = 200 yr range from 7,300 Gt to 29,600 Gt, with only about 25% of the entire ensemble successfully predicting the onset of accelerated volume loss at t = 130 years.

As more observations are assimilated, the ensemble spread is reduced, and the results of the deterministic simulations more closely align with the reference simulation. After 5 years of assimilation, both the deterministic and ensemble forecast simulations accurately reproduce changes in ice volume up to t = 15 years before beginning to diverge from the reference trajectory, resulting in 3,800 Gt of difference in volume loss by the end of the forecast period. Extending the assimilation period to 15 years reduces this discrepancy, with the deterministic forecast showing a smaller difference of 350 Gt in volume loss at t = 200 years. When the assimilation period is extended to 30 years, the agreement with the reference simulation improves even further, reducing the final volume loss difference to just 90 Gt. These results demonstrate that assimilating more observations not only improves agreement during the early forecast period but also enhances the accuracy of long-term

**Figure 8.** (a) Reference ice thickness (i.e., true) at t = 0 yr, (b) difference between true ice thickness and the ensemble mean ice thickness before assimilation (true - ensemble mean), (c)-(e) difference between true ice thickness and the corresponding ensemble mean ice thickness at (c) 5 years, (d) 15 years and (e) 30 years after assimilation. The localization radius is set to 4 km and the inflation factor is 1.12 with the ensemble size of 50. The green lines show the grounding line positions.

projections. With 15 years of assimilation, the ensemble spread decreases by approximately 86 % compared to the case without assimilation. Extending the assimilation window to 30 years results in little additional reduction in ensemble spread beyond what is achieved with 15 years of assimilation.

# 3.2 Results for Observing System Simulation Experiments (OSSEs)

In the context of our OSSEs, we evaluate the impact of varying cross-track spacings and grid resolutions of surface elevation data on the performance of DA in estimating the model state and parameters. Since the simulated surface elevation observations use different cross-track spacings and grid resolutions, we conduct sensitivity tests with an ensemble size of 50 to optimize both localization and inflation factors. When assimilating along-track surface elevations with 5 km and 10 km across track spacing, the best DA results are achieved with a localization radius of 4 km and the inflation between 1.10 and 1.14 for all variables (Fig. 10), similar to the DA results with full coverage of elevation data at each model mesh node in the twin experiment. As the across-track spacing increases to 10 and 15 km, the overall DA performance declines, indicated by an increase in the mean RMSE by up to 16 % for three estimated variables.

**Figure 9.** Changes in ice volume from ensemble forecast simulations with (a) no assimilation, (b) assimilation up to 5 years, (c) assimilation up to 15 years, and (d) assimilation up to 30 years. The red line shows the reference simulation, and the blue line shows the deterministic forecast simulation with the mean ensemble state. The gray lines show the forecast simulation of each ensemble member, and the dotted lines indicate the mean of ensemble simulations.

For the gridded elevation data with 1 km resolution, the optimal localization and inflation factors are 4 km and 1.12, respectively, for all variables. In experiments with gridded elevation data of 10 km and 20 km resolutions, the overall DA performance declines (i.e., an increase in RMSE) over a range of localization and inflation factors (Fig. 11). We find the minimum RMSE values at the end of the assimilation window with a localization radius of 6 - 8 km and inflation values of 1.02 - 1.06 for both 10 km and 20 km resolution data. While tuning these parameters helps improve performance, the overall accuracy remains lower than that achieved with 1 km grid data.




With the optimal parameter combinations identified for each elevation data type experiment, we conduct additional experiments exploring the impact of the prescribed uncertainty ( $\sigma_h$ ) of surface elevation data. To evaluate the DA performance, we summarized the RMSEs at the end of the assimilation window (at year 30) for each experiment in Table 2 and 3. The evolution of RMSEs over the assimilation period using the ground track and grid elevation observations are shown in Figs. 12 and 13, respectively.

When assimilating observations with 5 km across-track spacing and the same observational error as in the twin experiments ( $\sigma_h = 5 m$  and  $\sigma_v = 10 m/yr$ ), the DA performance, as measured by RMSEs, is comparable to that observed in the twin experiment (Table 2 and Fig. 12). As the across-track spacing of observed surface elevation increases, DA performance declines as expected. When assimilating data at 10 km or 15 km across-track spacing, RMSE values remain higher than those with 5 km spacing at t = 30 years, although RMSE values continue to decrease until the end of the assimilation window. A similar

Figure 10. Analysis ensemble mean RMSEs at t = 30 years as a function of the inflation factor and the localization radius for different across-track spacing of elevation data for (a-c) friction coefficient, (d-f) bed elevation, and (g-i) ice thickness. The grey shading indicates experiments that diverge by t = 30 yrs. The black box in each panel represents the minimum RMSE for each configuration.

result is observed with gridded elevation observations: high-resolution data (1 km) produces DA performance comparable to that of the twin experiment (Table 3 and Fig. 13). However, as the spatial resolution increases to 10 km and 20 km, the overall DA performance declines. For the 10 km grid data, only marginal improvements in the parameter and model state estimates are observed after 10 - 15 years of assimilation, while for the 20 km grid data, DA performance begins to degrade after 20 years of assimilation.


With 5 km across-track spacing, DA performance in retrieving bed topography and ice thickness decreases as the uncertainty in the surface elevation increases, both during the assimilation period (Fig. 12a,d,g) and at the end of the assimilation window (Table 2). DA performance for the friction coefficient shows little sensitivity to changes in elevation uncertainty, with RMSE\_C varying by only  $\sim$ 3 %, compared to  $\sim$  10 % variation in RMSE\_B and RMSE\_H at the end of the assimilation period. With the 10 km across-track data, DA performance for bed topography and ice thickness becomes more consistent across all uncertainty

**Figure 11.** Same as Fig. 10 but for different grid resolution of elevation data.


levels in elevation data, compared to the 5 km case (Fig. 12b,e,h). When using the 15 km across-track data, only surface elevation with an observational error standard deviation of 5 m improves bed and ice thickness estimation up to t = 30 years, while prescribed standard deviations of 10 - 20 m do not yield further improvements beyond 15 - 20 years of DA, and some increase in RMSE values is observed (Fig. 12c,f,i). During the assimilation period, the performance for bed topography and ice thickness is more similar across all uncertainty levels, compared to using the 5 km across-track data.

With the 1 km gridded elevation data, increasing uncertainty levels reduce the accuracy of bed and ice thickness estimation, while the friction coefficient does not show a clear pattern with varying uncertainty in surface elevation (13a,d,g). With coarser grid data (10 km and 20 km), the DA performance for all three variables shows less variation across different uncertainty levels during the assimilation window, compared to the 1 km grid data (13b,e,h and c,f,i).

**Table 2.** List of experiments using various across-track surface observations and analysis mean RMSEs t = 30 years.

| Experiment Name                                                               | RMSE_C ( $Pa  m^{-1/3} a^{1/3}$ ) | RMSE_B (m) | RMSE_H (m) |
|-------------------------------------------------------------------------------|-----------------------------------|------------|------------|
| Twin experiment ( $\sigma_h = 5 \text{ m}$ and $\sigma_v = 10 \text{ m/yr}$ ) | 296.01                            | 47.63      | 46.87      |
| Track_5km_ $\sigma_h$ _5_ $\sigma_v$ _10                                      | 306.89                            | 49.06      | 47.77      |
| Track_5km_ $\sigma_h$ _10_ $\sigma_v$ _10                                     | 304.96                            | 50.65      | 48.96      |
| Track_5km_ $\sigma_h$ _15_ $\sigma_v$ _10                                     | 305.62                            | 51.71      | 50.14      |
| Track_5km_ $\sigma_h$ _20_ $\sigma_v$ _10                                     | 313.61                            | 54.02      | 52.56      |
| Track_10km_ $\sigma_h$ _5_ $\sigma_v$ _10                                     | 338.28                            | 53.69      | 51.18      |
| ${\rm Track\_10km\_}\sigma_h\_{\rm 10\_}\sigma_v\_{\rm 10}$                   | 335.26                            | 52.84      | 50.62      |
| ${\rm Track\_10km\_}\sigma_h\_{\rm 15\_}\sigma_v\_{\rm 10}$                   | 350.69                            | 56.86      | 53.19      |
| Track_ $10$ km_ $\sigma_h$ _ $20$ _ $\sigma_v$ _ $10$                         | 341.78                            | 56.45      | 54.17      |
| Track_15km_ $\sigma_h$ _5_ $\sigma_v$ _10                                     | 410.10                            | 62.79      | 59.59      |
| ${\rm Track\_15km\_}\sigma_h\_{10\_}\sigma_v\_{10}$                           | 429.70                            | 73.43      | 70.03      |
| Track_15km_ $\sigma_h$ _15_ $\sigma_v$ _10                                    | 414.05                            | 72.55      | 65.96      |
| Track_15km_ $\sigma_h$ _20_ $\sigma_v$ _10                                    | 389.90                            | 69.62      | 65.74      |

# 4 Discussion




In this study, we show that the EAKF can effectively estimate both model state and parameter estimates for a semi-idealized glacier, especially in fast-flowing regions (e.g., velocity larger than 100 m/yr), which corresponds to regions around the grounding line, where the signal-to-noise ratio of velocity is relatively high. These results are consistent with those from previous studies (Gillet-Chaulet, 2020; Bonan et al., 2014, 2017), yet our approach employs a 2D model with unstructured meshes, enhancing its applicability to larger-scale ice sheet modeling simulations. Similar to earlier studies, assimilating new observations over the first few years significantly improves the accuracy of bed topography, friction coefficient, and ice thickness estimates in fast-flowing regions. A temporal decline in DA performance is observed during the assimilation period, likely due to a temporary mismatch between the model forecast and the observations, potentially caused by nonlinearities in the response to assimilated observations. As the assimilation continues, the filter gradually corrects these discrepancies, which leads to a subsequent reduction in RMSE. These fluctuations are not uncommon in ensemble data assimilation systems, especially in complex, nonlinear models where localized error growth can temporarily degrade performance (Carrassi et al., 2018). Although the slow-flowing regions—where the relative error in velocity observation is higher than in fast-flowing regions—show only limited improvements in basal conditions compared to the fast-flowing region, they still show notable improvements up to 300 km inland from the initial grounding lines (x = 150 km). These improvements allow more accurate forecasts of ice volume

**Table 3.** List of experiments using various gridded surface observations and analysis mean RMSEs at t = 30 years.

| Experiment Name                                                               | RMSE_C ( $Pa  m^{-1/3} a^{1/3}$ ) | RMSE_B (m) | RMSE_H (m) |
|-------------------------------------------------------------------------------|-----------------------------------|------------|------------|
| Twin experiment ( $\sigma_h = 5 \text{ m}$ and $\sigma_v = 10 \text{ m/yr}$ ) | 296.01                            | 47.63      | 46.87      |
| Grid_1km_ $\sigma_h$ _5_ $\sigma_v$ _10                                       | 291.38                            | 48.65      | 46.81      |
| $Grid_1km_\sigma_h_10_\sigma_v_10$                                            | 288.54                            | 48.62      | 47.43      |
| $Grid_1km_\sigma_{h}_15_\sigma_{v}_10$                                        | 291.29                            | 53.89      | 53.14      |
| Grid_1km_ $\sigma_h$ _20_ $\sigma_v$ _10                                      | 290.66                            | 54.88      | 53.97      |
| Grid_10km_ $\sigma_h$ _5_ $\sigma_v$ _10                                      | 437.48                            | 67.58      | 63.72      |
| $\mathrm{Grid}\_10\mathrm{km}\_\sigma_h\_10\_\sigma_v\_10$                    | 423.84                            | 66.76      | 63.99      |
| $\text{Grid}\_10\text{km}\_\sigma_h\_15\_\sigma_v\_10$                        | 430.58                            | 65.96      | 62.63      |
| Grid_10km_ $\sigma_h$ _20_ $\sigma_v$ _10                                     | 427.20                            | 66.61      | 63.20      |
| Grid_20km_ $\sigma_h$ _5_ $\sigma_v$ _10                                      | 432.50                            | 80.07      | 80.76      |
| $\mathrm{Grid}\_20\mathrm{km}\_\sigma_h\_10\_\sigma_v\_10$                    | 433.64                            | 80.69      | 79.96      |
| $\mathrm{Grid}\_20\mathrm{km}\_\sigma_h\_15\_\sigma_v\_10$                    | 431.91                            | 77.42      | 78.97      |
| Grid_20km_ $\sigma_h$ _20_ $\sigma_v$ _10                                     | 433.06                            | 77.84      | 79.39      |

loss for up to 200 years, as the grounding line retreats by approximately 150 km (to x = 300 km) by the end of the reference simulation.

For the initial estimates of the model parameters—bed topography and friction coefficient—we assume reasonably accurate prior knowledge of initial conditions and prescribe covariance models to establish spatial correlation within each parameter. In real glacier applications, however, these assumptions may not hold. For better DA results, more accurate measurements and/or prior information for bed conditions are required, such as additional radar measurements of bed topography and potential relationships between geophysical observations (e.g., seismic or radar-based measures) and friction (Kyrke-Smith et al., 2017; Haris et al., 2024). Alternatively, multi-model reconstructions of parameters could be leveraged to generate initial ensembles of parameters and determine the ensemble spread (Gillet-Chaulet, 2020). Our DA results, along with localization and inflation factors, may depend on assumptions about how the initial ensemble is generated. Exploring how gaps in prior information affect DA results could provide valuable insights, particularly in understanding the robustness of DA results when challenged with realistic data limitations and parameter uncertainties.

The robust performance of the EAKF in constraining the basal conditions and initial ice sheet state for future projection has been achieved with the ensemble size of 30, the smallest explored in this study, consistent with previous studies performing data assimilation for flowline models (Bonan et al., 2014; Gillet-Chaulet, 2020). We further show that increasing the ensemble size allows robust DA performance over a wider range of localization radii and inflation factors and produces only marginally improved performance in retrieving basal conditions with shorter assimilation windows. Therefore, a majority of experiments

**Figure 12.** The evolution of ensemble mean RMSEs for (a-c) friction coefficient, (d-f) bed topography, and (g-i) ice thickness under different across-track spacings of surface elevation observations and varying levels of surface elevation uncertainty.

performed in this study use an ensemble size of only 50 members, which we find to be a reasonable tradeoff between data assimilation accuracy and computational efficiency.

Larger ensemble sizes could improve data assimilation performance but may also introduce challenges that must be carefully managed, particularly in long assimilation periods or highly nonlinear systems, as in this study. In our experiments, it is possible that the inflation and localization parameters used for the 100-member ensemble were not optimal for later assimilation periods, leading to slightly degraded performance after year 15. This suggests that filter performance does not necessarily scale linearly with ensemble size and highlights the importance of adaptive inflation/localization techniques or diagnostics for dynamically adjusting filter settings.



In this study, we use spatially and temporally uniform inflation and localization techniques to stabilize the filter, similar to previous studies (Bonan et al., 2014; Gillet-Chaulet, 2020). The optimal inflation factors for this study (1.10 - 1.18) are similar to values (0.98 - 1.14) from earlier studies (Bonan et al., 2014; Gillet-Chaulet, 2020). For localization radius, the best results were obtained with a radius of 4 - 8 km. Choosing too small of a radius causes the EAKF to underestimate spatial error correlations and diverge with time. In our experiments, this is evident when the localization radius falls below the specific threshold of each variable (e.g., 4 km for friction and 6 km for bed topography).

The optimal localization radius found in this study compares to previous flowline model studies that suggested optimal localization radii of 4 - 16 km for a grid size of 0.2 km (Gillet-Chaulet, 2020) and 80 - 120 km for a grid size of 5 km (Bonan

**Figure 13.** Same as Fig 13, but using different grid resolutions for surface elevation observations.


et al., 2014). The differences in the optimal localization radius likely comes from the differences in model configuration, dimensionality, and spatial resolution. Our study uses a 2D unstructured mesh with relatively fine spatial resolution, whereas previous studies using flowline models with coarser grids may require broader localization to account for longer correlation length scales. The localization radius is determined through a set of sensitivity experiments and is based on the expected spatial correlation length scale of the parameters, which may depend on the size of flow features or stress balance regimes. Given our use of a 2D unstructured mesh, adaptive inflation (El Gharamti, 2018) and localization (Bishop and Hodyss, 2007) can be viable alternatives, as each node has a different number of observations to be assimilated.

In our twin experiment and projections, we find that assimilating more observations years to estimate basal conditions improves the accuracy of model projections with reduced uncertainty through the corresponding projection period. Without data assimilation, individual ensemble members show a large spread of future projections due to nonlinear feedbacks triggered by small deviations from the true basal field. While the deterministic forecast, initialized with the ensemble mean of the basal fields, captures the overall trend in ice volume change from the reference simulation, reducing local extremes, it still yields consistent discrepancies throughout the assimilation period. Assimilating surface observations for up to 15 years results in ensemble and deterministic ice volume loss forecasts that closely match the reference simulation for up to 100 years, with much reduced ensemble spread and ice volume loss difference limited to approximately 300 Gt (compared to ~2000 Gt with no assimilation). Extending the assimilation window to 30 years allows forecast simulations to match the reference simulation for up to 200 years. Notably, the 200-year reference simulation includes a phase of accelerated volume loss after 130 years, which

may represent a plausible sea level rise scenario for the coming century. Our results suggest that assimilating observations even before such nonlinear transitions can still reproduce accurate long-term projections—provided that the model state and parameters are well constrained. Our projections further show a better match to the reference simulations compared to those from a previous study (Gillet-Chaulet, 2020), potentially due to our use of more observations with smaller error variance ( $\sigma_v$  and  $\sigma_h$ ). The method used here, which assimilates time series of observations, maintains consistency with transient changes in the model state and provides an optimal initial condition for changing glaciers.







In this study, we focus on estimating two constant-in-time parameter fields and the model state using annual observations over assimilation windows of varying lengths (5, 15, and 30 years). This choice is motivated by both the timescales associated with glacier dynamics and the current capabilities of observing platforms. However, the relative importance of the assimilation window length (i.e., total time span) versus the number of assimilation cycles (i.e., update frequency) remains an open question. To explore this, we conduct an additional experiment using semiannual observations under the same setup as the twin experiment (Fig. A1). The results suggest that semiannual observations lead to a faster reduction in RMSE for both the model state and parameters. However, the improvement at the end of the 30-year assimilation window, compared to annual assimilation, remains limited. This limited benefit is likely due to the nature of the parameters and state variables considered in this study—constant-in-time fields and annual-scale variability—which allow sufficient information to accumulate over time for a fixed target. Once sufficient assimilation cycles have passed, the parameters become well constrained, and more frequent updates offer little additional improvement. These findings suggest that, for slowly varying or static variables, increasing observation frequency can accelerate convergence toward the true state and parameter values, but may not results in additional improvement beyond a certain number of assimilation cycles. In contrast, if parameters or states change more rapidly or nonlinearly, a longer assimilation window or more complex update schemes might be needed to achieve similar improvements. Future work should explore the sensitivity of EnKF performance to both assimilation frequency and window length to identify optimal configurations for real glacier systems with time-varying parameters and limited observation periods.

The purpose of our OSSEs—which use synthetic observations to evaluate the potential benefits of different observing strategies—is to demonstrate their capabilities within the ISSM–EnKF framework. Our OSSE experiments show that an EnKF can effectively assimilate various types of surface elevation observations (both grid and track) to evaluate the impact of different observational products. We find that higher spatial resolution in elevation observations substantially improves DA performance. For example, gridded data at 1 km resolution and track-based data with 5 km across-track spacing yield results comparable to those in our twin experiment with full coverage. In principle, higher-resolution data (e.g., 100 m) could further improve data assimilation performance by providing finer spatial detail on surface features and more precise constraints on model parameters. However, the benefit of finer resolution may decrease beyond a certain threshold due to increased observational noise, modeling uncertainties, and the inherent spatial correlation scale of the parameters being estimated. In contrast, lower-resolution datasets—such as 10–20 km gridded data or 15 km track spacing—lead to a noticeable decline in DA performance. In these cases, the filter struggles to resolve finer-scale variations in the ice sheet state, resulting in larger RMSE values. Additionally, the marginal improvements or increases in RMSE observed at coarser resolutions after 10–15 years suggest that, beyond a certain spatial threshold, additional data points do not improve—may even degrade—long-term parameter and the model

state estimations (Fig. 12 and 13). These results highlight the importance of balancing observational density and coverage to maximize DA performance over the historical period.

The OSSE experiments also provide a basic demonstration of the impact of observational error on DA performance, with particular benefits of lower surface elevation uncertainties on bed topography and ice thickness estimation when using high-resolution data. These benefits become less pronounced when lower-resolution elevation data are used. In contrast, friction coefficient retrieval shows no clear pattern in response to the prescribed surface elevation uncertainty, regardless of the data resolution. However, when we vary velocity observations errors while keeping elevation uncertainty constant, we observe that reducing velocity errors improves the estimation of the friction coefficient (Fig. A2), as well as bed topography and ice thickness estimates. Given our semi-idealized model domain and simplified error propagation method, we do not derive specific error thresholds for effective ice sheet model parameters and state estimation. However, we note that a proper specification of observation uncertainty is likely critical for accurate DA performance, and the relative importance of velocity versus elevation uncertainty depends on the specific variable being estimated.

Despite the promising results demonstrated in this study, several limitations exist that must be acknowledged and addressed in future research. First, our study utilizes yearly synthetic observations with uniformly homogeneous error variance, which do not fully capture the complexities and variability present in real observations. In addition, we assume full spatial and temporal coverage of velocity data to isolate and focus on the impact of surface elevation observations. While this simplifies the analysis, it is an idealized scenario; future research should explore more realistic data configurations, including partial velocity coverage, and assess the trade-offs between observation density, cost, and assimilation performance. A joint analysis of surface and velocity observations would provide a more robust understanding of their relative contributions to improving model estimates. Future research should also consider more sophisticated methods to account for observations from diverse sensors, coverage, varying periods, state dependence, and collection frequencies, as well as their associated error covariance matrices. This includes conducting more comprehensive OSSEs with a broader range of potential observations.

Additionally, this study focused on only one filter algorithm with a limited range of inflation and localization factors, which may not adequately explore the full potential and scalability of the DA method. Future studies should investigate different types of filter algorithms and a variety of inflation and localization techniques to better optimize the assimilation process for ice sheet modeling. Furthermore, incorporating more comprehensive climate processes could enhance the predictive capabilities of our simulations. For example, integrating the firn process into the model could help not only in accurately modeling the grounding line position (Gillet-Chaulet, 2020) but also in properly determining observation errors in the DA process.

Although ensemble-based data assimilation offers conceptual and practical advantages, its computational cost is often considered a limiting factor. In this study, we did not perform a direct computational comparison between ensemble and variational (transient) DA approaches. Such a comparison is challenging due to their fundamentally different implementations. For example, variational DA in ISSM relies on automatic differentiation (AD), which can be memory-intensive, whereas ensemble DA increases computational cost primarily by requiring multiple forward simulations. However, ensemble approaches can be parallelized, as each ensemble member's forward run can be distributed across separate cores or nodes, and the DA process

here is managed through DART, which supports parallel computing. While formal benchmarking was beyond the scope of this study, it would be valuable in future work to quantify computational trade-offs across DA methods in ice sheet modeling.

Our experimental design also assumed perfect knowledge of all model parameters except for basal friction and bed topography. This choice was made to facilitate learning about the DA system in a controlled setting and to keep the experimental setup more tractable, while also allowing for direct comparison with Gillet-Chaulet (2020). However, this approach limits the realism of experiments. In practice, parameters such as ice viscosity and climate forcing are also poorly constrained and may vary in both space and time. For example, uncertainties in viscosity may interact with basal friction during assimilation, potentially leading to parameter compensation effects. Future sensitivity studies should explore how mis-specified background parameters (e.g., biased viscosity fields) affect the estimation of other parameters and whether such compensation leads to biased or unstable forecasts. Although this study focuses on estimating two constant-in-time parameter fields (friction coefficient and bed topography), the DART–ISSM framework is well-suited for the joint estimation of multiple spatially or temporally varying parameters. Extending the current configuration to include additional unknowns—such as ice viscosity, accumulation rate, or time-varying boundary conditions—represents a valuable next step toward more realistic data assimilation in ice sheet modeling.

### 5 Conclusions







In this study, we introduce an ensemble Kalman filter-based data assimilation (DA) framework to calibrate a 2D plan-view ice model. Using a synthetic twin experiment, we showed that the ensemble DA method effectively recovers basal conditions (friction coefficient and bed topography) and ice thickness after several assimilation cycles. While a temporal decline in DA performance is observed during the assimilation period—likely due to model nonlinearity—assimilating more observations generally improves the accuracy of the model state and parameters. With 30 years of assimilated surface observations, the deterministic forecast reproduces the total ice volume change of the reference simulation within approximately 1% over a 200year period. We also conduct Observing System Simulation Experiments (OSSEs) using the same model domain as the twin experiment but with synthetic elevation observations along ground track and gridded data that emulate the ICESat-2 ATL11 and ATL15 products, respectively. These experiments present the potential surface elevation product that can be used to accurately estimate bed conditions and the model state of the idealized glacier. The results highlight the crucial role of spatial resolution of surface elevation data in the DA performance. In addition, we find that varying levels of observational uncertainty—not necessarily smaller—can lead to improved assimilation outcomes, which highlights the importance of a more accurate representation of observation uncertainty in the DA process. The ensemble DA framework, which assimilates observations from multiple time points, holds significant potential for application to real glaciers to better estimate the current and future changes in ice sheet state variables. This framework also provides advantages for OSSEs aimed at testing various observational settings, as it requires less numerical effort than variational methods that assimilate time series of observations, making it a practical and effective tool in ice sheet modeling.

Code and data availability. The ISSM is open source and the source code of ISSM is available at https://github.com/ISSMteam/ISSM. The source code of DART is available at https://github.com/NCAR/DART (DART, 2024). The script for the results and figures are available at https://doi.org/10.5281/zenodo.14722078.

# Appendix A


We conduct additional experiments to assess the impact of assimilation frequency and different levels of velocity uncertainty on the results. We use full spatial coverage of surface velocity and elevation data—as in the twin experiments—to avoid the influence of spatial data gaps (e.g., in surface elevation).

**Figure A1.** The evolution of mean analysis RMSE for (a) friction coefficient, (b) bed topography, and (c) ice thickness, using full spatial coverage of surface velocity and elevation data (as in the twin experiments), under different assimilation frequencies (blue: annual, red: semiannual)

Figure A2. The evolution of mean analysis RMSE for (a) friction coefficient, (b) bed topography, and (c) ice thickness, using full spatial coverage of surface velocity and elevation data (as in the twin experiments), under different levels of uncertainty in surface velocity observations.

*Author contributions.* YC designed the experiments and conducted all simulations with help from AP, DF and JP. YC drafted the initial manuscript with inputs from AP, and all authors contributed to editing the manuscript.

Competing interests. The contact author has declared that none of the authors has any competing interests.

*Acknowledgements.* This work was supported by NASA under a Decadal Survey Incubation (DSI) Surface Topography and Vegetation team award (#80NSSC22K1112) and Cryospheric Science Program (#80NSSC24K1461).

- Anderson, J., Hoar, T., Raeder, K., Liu, H., Collins, N., Torn, R., and Avellano, A.: The data assimilation research testbed: A community facility, Bull. Am. Meteorol. Soc., 90, 1283-1296, 2009.
- Anderson, J. L.: An ensemble adjustment Kalman filter for data assimilation, Mon. Weather Rev., 129, 2884–2903, 2001.
- Anderson, J. L.: An adaptive covariance inflation error correction algorithm for ensemble filters, Tellus A: Dynamic meteorology and 565 oceanography, 59, 210-224, 2007.
  - Arnold Jr, C. P. and Dey, C. H.: Observing-systems simulation experiments: Past, present, and future, Bull. Am. Meteorol. Soc., 67, 687–695, 1986.
  - Asay-Davis, X. S., Cornford, S. L., Durand, G., Galton-Fenzi, B. K., Gladstone, R. M., Gudmundsson, G. H., Hattermann, T., Holland, D. M., Holland, D., Holland, P. R., et al.: Experimental design for three interrelated marine ice sheet and ocean model intercomparison projects; MISMIP v. 3 (MISMIP+), ISOMIP v. 2 (ISOMIP+) and MISOMIP v. 1 (MISOMIP1), Geosci, Model Dev., 9, 2471–2497, 2016.
  - Bishop, C. H. and Hodyss, D.: Flow-adaptive moderation of spurious ensemble correlations and its use in ensemble-based data assimilation, O. J. R. Meteorol, Socv., 133, 2029–2044, 2007.
    - Bonan, B., Nodet, M., Ritz, C., and Peyaud, V.: An ETKF approach for initial state and parameter estimation in ice sheet modelling, Nonlinear Process. Geophys., 21, 569-582, 2014.
- Bonan, B., Nichols, N. K., Baines, M. J., and Partridge, D.: Data assimilation for moving mesh methods with an application to ice sheet modelling, Nonlinear Process. Geophys., 24, 515–534, 2017.
  - Boukabara, S.-A., Moradi, I., Atlas, R., Casey, S. P., Cucurull, L., Hoffman, R. N., Ide, K., Kumar, V. K., Li, R., Li, Z., et al.: Community global observing system simulation experiment (OSSE) package (CGOP): description and usage, J. Atmos. Ocean, Technol., 33, 1759– 1777, 2016.
- Brondex, J., Gillet-Chaulet, F., and Gagliardini, O.: Sensitivity of centennial mass loss projections of the Amundsen basin to the friction law, 580 Cryosphere, 13, 177-195, 2019.
  - Carrassi, A., Bocquet, M., Bertino, L., and Evensen, G.: Data assimilation in the geosciences: An overview of methods, issues, and perspectives, Wiley Interdiscip. Rev. Clim. Change, 9, e535, 2018.
- Choi, Y., Seroussi, H., Morlighem, M., Schlegel, N.-J., and Gardner, A.: Impact of time-dependent data assimilation on ice flow model 585 initialization and projections: a case study of Kier Glacier, Greenland, Cryosphere, 17, 5499–5517, 2023.
  - Cook, S., Gillet-Chaulet, F., and Fürst, J.: Robust reconstruction of glacier beds using transient 2D assimilation with Stokes, J. Glaciol., 69, 1393-1402, 2023.
  - Dai, C. and Howat, I. M.: Measuring lava flows with ArcticDEM: Application to the 2012–2013 eruption of Tolbachik, Kamchatka, Geophys. Res. Lett., 44, 12-133, 2017.
- DART: The Data Assimilation Research Testbed (Version 11.8.0) [Software], https://doi.org/10.5065/D6WQ0202, 2024.
  - Dowdeswell, J. A. and Evans, S.: Investigations of the form and flow of ice sheets and glaciers using radio-echo sounding, Rep. Prog. Phys., 67, 1821, 2004.
  - El Gharamti, M.: Enhanced adaptive inflation algorithm for ensemble filters, Mon. Wea. Rev., 146, 623-640, 2018.
  - Evans, S. and Robin, G. d. O.: Glacier depth-sounding from air, Nature, 210, 883–885, 1966.
- Favier, L., Durand, G., Cornford, S. L., Gudmundsson, G. H., Gagliardini, O., Gillet-Chaulet, F., Zwinger, T., Payne, A. J., and Le Brocq, A. M.: Retreat of Pine Island Glacier controlled by marine ice-sheet instability, Nat. Clim. Change, 4, 117–121, 2014.

- Fournier, A., Fussell, D., and Carpenter, L.: Computer rendering of stochastic models, Commun. ACM, 25, 371–384, 1982.
- Gaspari, G. and Cohn, S. E.: Construction of correlation functions in two and three dimensions, Q. J. R. Meteorol. Soc., 125, 723–757, 1999.
- Gillet-Chaulet, F.: Assimilation of surface observations in a transient marine ice sheet model using an ensemble Kalman filter, Cryosphere, 14, 811–832, 2020.
  - Gillet-Chaulet, F., Gagliardini, O., Seddik, H., Nodet, M., Durand, G., Ritz, C., Zwinger, T., Greve, R., and Vaughan, D. G.: Greenland Ice Sheet contribution to sea-level rise from a new-generation ice-sheet model, Cryosphere, 6, 1561–1576, 2012.
  - Glen, J. W.: The creep of polycrystalline ice, Proc. R. Soc. A, 228, 519-538, 1955.

- Goelzer, H., Nowicki, S., Payne, A., Larour, E., Seroussi, H., Lipscomb, W. H., Gregory, J., Abe-Ouchi, A., Shepherd, A., Simon, E., Agosta,
- C., Alexander, P., Aschwanden, A., Barthel, A., Calov, R., Chambers, C., Choi, Y., Cuzzone, J., C., D., Edwards, T., Felikson, D., Fettweis, X., Golledge, N. R., Greve, R., Humbert, A., Huybrechts, P., Le clec'h, S., Lee, V., Leguy, G., Little, C., Lowry, D. P., Morlighem, M., Nias, I., Quiquet, A., Rückamp, M., Schlegel, N.-J.and Slater, D. A., Smith, R. S., Straneo, F., Tarasov, L., van de Wal, R., and van den Broeke, M.: The future sea-level contribution of the Greenland ice sheet: a multi-model ensemble study of ISMIP6, Cryosphere, 2020.
- Goldberg, D. N. and Heimbach, P.: Parameter and state estimation with a time-dependent adjoint marine ice sheet model, Cryosphere, 17, 1659–1678, 2013.
  - Goldberg, D. N., Heimbach, P., Joughin, I., and Smith, B.: Committed retreat of Smith, Pope, and Kohler Glaciers over the next 30 years inferred by transient model calibration, Cryosphere, 9, 2429–2446, 2015.
  - Haris, R., Chu, W., and Robel, A.: What can radar-based measures of subglacial hydrology tell us about basal shear stress? A case study at Thwaites Glacier, West Antarctica, J. Glaciol., pp. 1–9, 2024.
- Hoffman, R. N. and Atlas, R.: Future observing system simulation experiments, Bull. Am. Meteorol. Soc., 97, 1601–1616, 2016.
  - Iglesias, M. A., Law, K. J., and Stuart, A. M.: Ensemble Kalman methods for inverse problems, Inverse Probl., 29, 045 001, 2013.
  - Intergovernmental Panel on Climate Change (IPCC): Ocean, Cryosphere and Sea Level Change, pp. 1211–1362, Cambridge University Press, 2023.
- Kyrke-Smith, T. M., Gudmundsson, G. H., and Farrell, P. E.: Can seismic observations of bed conditions on ice streams help constrain parameters in ice flow models?, J. Geophys. Res. Earth Surf., 122, 2269–2282, 2017.
  - Larour, E., Schiermeier, J., Rignot, E., Seroussi, H., Morlighem, M., and Paden, J.: Sensitivity Analysis of Pine Island Glacier ice flow using ISSM and DAKOTA, J. Geophys. Res., 117, F02009, 1–16, 2012.
  - Larour, E., Utke, J., Csatho, B., Schenk, A., Seroussi, H., Morlighem, M., Rignot, E., Schlegel, N., and Khazendar, A.: Inferred basal friction and surface mass balance of the Northeast Greenland Ice Stream using data assimilation of ICESat (Ice Cloud and land Elevation Satellite) surface altimetry and ISSM (Ice Sheet System Model), Cryosphere, 8, 2335–2351, 2014.
  - MacAyeal, D. R.: Large-scale ice flow over a viscous basal sediment: Theory and application to Ice Stream B, Antarctica, J. Geophys. Res., 94, 4071–4087, 1989.
  - MacAyeal, D. R.: The basal stress distribution of Ice Stream E, Antarctica, Inferred by Control Methods, J. Geophys. Res., 97, 595–603, 1992.
- Masutani, M., Woollen, J. S., Lord, S. J., Emmitt, G. D., Kleespies, T. J., Wood, S. A., Greco, S., Sun, H., Terry, J., Kapoor, V., et al.:

  Observing system simulation experiments at the National Centers for Environmental Prediction, J. Geophys. Res. Atmos., 115, 2010.
  - Morlighem, M., Rignot, E., Seroussi, H., Larour, E., Ben Dhia, H., and Aubry, D.: Spatial patterns of basal drag inferred using control methods from a full-Stokes and simpler models for Pine Island Glacier, West Antarctica, Geophys. Res. Lett., 37, 1–6, https://doi.org/10.1029/2010GL043853, 2010.

- 635 Morlighem, M., Seroussi, H., Larour, E., and Rignot, E.: Inversion of basal friction in Antarctica using exact and incomplete adjoints of a higher-order model, J. Geophys. Res., 118, 1746–1753, 2013.
  - Morzfeld, M. and Hodyss, D.: A theory for why even simple covariance localization is so useful in ensemble data assimilation, Mon. Weather Rev., 151, 717–736, 2023.
- Mouginot, J., Rignot, E., Scheuchl, B., and Millan, R.: Comprehensive Annual Ice Sheet Velocity Mapping Using Landsat-8, Sentinel-1, and RADARSAT-2 Data, Remote Sens., 9, 2017.
  - Müller, S., Schüler, L., Zech, A., and Heße, F.: GSTools v1. 3: a toolbox for geostatistical modelling in Python, Geosci. Model Dev., 15, 3161–3182, 2022.
  - Nowicki, S. and Seroussi, H.: Projections of future sea level contributions from the Greenland and Antarctic Ice Sheets: Challenges beyond dynamical ice sheet modeling, Oceanography, 31, 2018.
- Nowicki, S. M. J., Payne, A. J., Larour, E., Seroussi, H., Goelzer, H., Lipscomb, W. H., Gregory, J., Abe-Ouchi, A., and Shepherd, A.: Ice Sheet Model Intercomparison Project (ISMIP6) contribution to CMIP6, Geosci. Model Dev., 9, 4521–4545, 2016.
  - Rodriguez-Morales, F., Gogineni, S., Leuschen, C. J., Paden, J. D., Li, J., Lewis, C. C., Panzer, B., Gomez-Garcia Alvestegui, D., Patel, A., Byers, K., Crowe, R., Player, K., Hale, R. D., Arnold, E. J., Smith, L., Gifford, C. M., Braaten, D., and Panton, C.: Advanced Multifrequency Radar Instrumentation for Polar Research, IEEE Trans. Geosc. and Rem. Sens., 52, 2824–2842, 2014.
- Seroussi, H., Morlighem, M., Rignot, E., Larour, E., Aubry, D., Ben Dhia, H., and Kristensen, S. S.: Ice flux divergence anomalies on 79north Glacier, Greenland, Geophys. Res. Lett., 38, 1–5, 2011.
  - Seroussi, H., Morlighem, M., Larour, E., Rignot, E., and Khazendar, A.: Hydrostatic grounding line parameterization in ice sheet models, Cryosphere, 8, 2075–2087, 2014.
- Seroussi, H., Nowicki, S., Simon, E., Abe-Ouchi, A., Albrecht, T., Brondex, J., Cornford, S., Dumas, C., Gillet-Chaulet, F., Goelzer, H.,

  Golledge, N. R., Gregory, J. M., Greve, R., Hoffman, M. J., Humbert, A., Huybrechts, P., Kleiner, T., Larour, E., Leguy, G., Lipscomb,
  W. H., Lowry, D., Mengel, M., Morlighem, M., Pattyn, F., Payne, A. J., Pollard, D., Price, S. F., Quiquet, A., Reerink, T. J., Reese, R.,

  Rodehacke, C. B., Schlegel, N.-J., Shepherd, A., Sun, S., Sutter, J., Van Breedam, J., van de Wal, R. S. W., Winkelmann, R., and Zhang,
  T.: initMIP-Antarctica: an ice sheet model initialization experiment of ISMIP6, Cryosphere, 13, 1441–1471, 2019.
- Seroussi, H., Nowicki, S., Payne, A. J., Goelzer, H., Lipscomb, W. H., Abe-Ouchi, A., Agosta, C., Albrecht, T., Asay-Davis, X., Barthel,
  A., Calov, R., Cullather, R., Dumas, C., Galton-Fenzi, B. K., Gladstone, R., Golledge, N. R., Gregory, J. M., Greve, R., Hattermann, T.,
  Hoffman, M. J., Humbert, A., Huybrechts, P., Jourdain, N. C., Kleiner, T., Larour, E., Leguy, G. R., Lowry, D. P., Little, C. M., Morlighem,
  M., Pattyn, F., Pelle, T., Price, S. F., Quiquet, A., Reese, R., Schlegel, N.-J., Shepherd, A., Simon, E., Smith, R. S., Straneo, F., Sun, S.,
  Trusel, L. D., Van Breedam, J., van de Wal, R. S. W., Winkelmann, R., Zhao, C., Zhang, T., and Zwinger, T.: ISMIP6 Antarctica: a
  multi-model ensemble of the Antarctic ice sheet evolution over the 21<sup>st</sup> century, The Cryosphere, 14, 3033–3070, 2020.
- Smith, B., Fricker, H. A., Holschuh, N., Gardner, A. S., Adusumilli, S., Brunt, K. M., Csatho, B., Harbeck, K., Huth, A., Neumann, T., et al.: Land ice height-retrieval algorithm for NASA's ICESat-2 photon-counting laser altimeter, Remote Sens. Environ., 233, 111 352, 2019.
  - Smith, B., Dickinson, S., Jelley, B. P., Neumann, T. A., Hancock, D., Lee, J., and Harbeck, K.: ATLAS/ICESat-2 L3B Slope-Corrected Land Ice Height Time Series, Version 6, https://doi.org/10.5067/ATLAS/ATL11.006, 2023.
- Smith, B., Sutterley, T., Dickinson, S., Jelley, B. P., Felikson, D., Neumann, T. A., Fricker, H. A., Gardner, A. S., Padman, L., Markus, T.,

  Kurtz, N., Bhardwaj, S., Hancock, D., and Lee, J.: ATLAS/ICESat-2 L3B Gridded Antarctic and Arctic Land Ice Height Change, Version
  4, https://doi.org/10.5067/ATLAS/ATL15.004, 2024.

- Tippett, M. K., Anderson, J. L., Bishop, C. H., Hamill, T. M., and Whitaker, J. S.: Ensemble square root filters, Mon. Weather Rev., 131, 1485–1490, 2003.
- Whitaker, J. S. and Hamill, T. M.: Ensemble data assimilation without perturbed observations, Monthly weather review, 130, 1913–1924, 2002.
  - Zhang, Y.-F., Bitz, C. M., Anderson, J. L., Collins, N., Hendricks, J., Hoar, T., Raeder, K., and Massonnet, F.: Insights on sea ice data assimilation from perfect model observing system simulation experiments, J. Clim., 31, 5911–5926, 2018.
  - Zubrow, A., Chen, L., and Kotamarthi, V.: EAKF-CMAQ: Introduction and evaluation of a data assimilation for CMAQ based on the ensemble adjustment Kalman filter, Journal of Geophysical Research: Atmospheres, 113, 2008.