# Peer review of "Estimation of the state and parameters in ice sheet model using an ensemble Kalman filter and Observing System Simulation Experiments"

_EGUsphere, 2025_

## Referee Comment (RC1)

**Review of "Estimation of the state and parameters in ice sheet model using an ensemble Kalman filter and Observing System Simulation Experiments"**

March 27, 2025

**General comments**

The manuscript by Choi et al. presents a data assimilation framework to improve the projection capabilities of ice sheet models. Specifically, the performance of an Ensemble Adjustment Kalman Filter in constraining the model state (ice thickness) and basal conditions (basal friction coefficient and bed topography) of a 2D plan-view ice model is assessed. Their results indicate that assimilating more observations generally increases the accuracy of model projections, with projections for up to 200 years in close agreement with the reference simulation. The performance of the data assimilation method is sensitive to the observational error as well as the cross-track spacing and grid resolution of surface elevation data.

I believe the science behind this study is sound and aligns with the focus of The Cryosphere (TC). However, the presentation of the methodology lacks clarity, at times adding avoidable confusion (e.g., the introduction of both acronyms EnKF and EAKF). This overall issue is addressed in more detail in the specific comments below, but I strongly suggest the addition of a flowchart outlining the methodology (ice sheet model and data assimilation) and experimental design (twin experiment and OSSEs). Furthermore, parts of the experimental design are currently placed within the results section and key aspects of the results are not addressed (e.g., why is the RMSE_C for Grid_20km_$\sigma_{\mathrm{h}}$_20_$\sigma_{\mathrm{v}}$_10 smaller than for the same grid resolution with smaller uncertainties as well as all 10 km grid resolution experiments?). Considering the performance of the data assimilation method is sensitive to the uncertainty in surface elevation observations, I believe also determining the effect of various uncertainties in the velocity data would add further value to the manuscript (perhaps as supplementary material). Finally, potential reasons/explanations for model results are often missing, e.g., why is the range of optimal localization radius (4 - 8 km) a lot smaller than suggested by previous studies (4 - 120 km)? I recommend the authors also take my specific comments listed below into account.

**Specific comments**

L25: This sentence is very similar to the second sentence in the introduction. Instead, consider opening with a sentence about the different DA methods (variational vs. methods leveraging time-varying observations). Then proceed to discuss advantages/disadvantages of each.

L28: Double brackets.

L33: Consider starting a new paragraph before *Alternatively*.

L37 – *assimilation period*: Readers unfamiliar with DA might not know what exactly you refer to here. It becomes a lot clearer later on, but it would be nice to have a brief definition here (similar for other DA-specific terms, e.g., *data denial experiments* in L61).

L40: Move further up to the rest of the discussion on variational methods.

L42: Consider introducing the term ensemble DA in general before describing the specific EnKF (e.g., ensemble DA vs. variational methods).

L54: Why are ensemble DA methods less commonly used in ice sheet modelling?

L63 – *(OSSEs)(OSSEs, ...)*.

L65 – *appropriate observation error distribution*: How do you determine if the distribution is appropriate or not?

L71: Although it is addressed in more detail in the next sentence, I believe adding *(ice thickness)* just after *model state* would add clarity.

L75 – *estimated state and parameters*: For consistency, I recommend using estimated *model* state and parameters throughout the manuscript.

L75 – *true reference values*: At this point in the manuscript, it is not clear what the *true reference values* are and how you obtain them.

L83-84: Remove this sentence and add the reference to the description of the specific sections in the text above.

L86-88: Repetition of the text just before the Methods section.

L101: I am not familiar with this specific method, but a standard deviation of 500 m seems quite large considering the bed varies only between $z_{b,deep} = -720$ m and $\sim500$ m (really difficult to see in Fig. 1a)

Fig. 1: I believe using two separate 2D plots instead of the 3D plot would make the identification of certain details and interpretation of the plot a lot easier. As you are already showing the bed topography in Fig. 2, I recommend combining Fig. 1 and 2 into a single plot with panels a: ice surface elevation, b: bed topography, and c: ice velocity. Note that the rainbow colour scheme is not in line with the journal guidelines. You can check all of your plots with the colour blindness simulator (https://www.color-blindness.com/coblis-color-blindness-simulator/). The fonts in panel b are too small and I recommend using a different colour for your contour lines.

Fig. 2: Why are you using such an asymmetrical (about y=40 km) bed topography compared to the commonly used symmetrical approach in idealized studies? The y-label in Fig. 2 indicates the domain ranges from 20 to 100 km, whereas in Fig. 1 it is 0 to 80 km. You also might want to remove the white margins at the top and bottom. How can the bed elevation be -1500 m when Eq. 1 limits the bed to $z_{b,deep} = -720$ m?

L103: Consider adding an additional panel (d) showing the triangular mesh to the new Fig. 1. Are you using adaptive mesh refinement, e.g., following the grounding line?

L104: I suggest adding another panel (e) for the basal friction coefficient to Fig. 1 or at least refer to Fig. 6a here.

Eq. 5: In case you are working in LaTeX, I recommend using $\left($ and $\right)$ to get brackets of the correct size.

Eq. 6: Same as for Eq. 5

L114: Is C in Eq. 7 different from the one described in Eq. 4? If not, then remove *C is a friction coefficient.*

L117: Remove *equal.*

L124: Do you consider a melt rate of 200 m/yr realistic given that maximum present-day melt rates are around 100 m/yr?

L125-127: This belongs into results.

L129: Sec. 2.3 and 2.4 are referenced before 2.2.

L134: modified version of the *Ensemble* Kalman Filter? As I mentioned above, using EnKF and EAKF is confusing, especially since EAKF is introduced but only used within this paragraph.

L136-143: This is where I think a flowchart would really help the reader to follow the details of your method. Ideally, the flowchart should outline the details of the EAKF and how it relates to your specific study. For example: How do ensemble members differ? What exactly is your model forecasting? How is the observation window specified? Which ice sheet variables are considered in the state vector? What are the state variables?

L142: How would adding extra variables, like surface velocity, to the state vector affect your results?

L143-144: EnKFs or EAKFs? Does this challenge arise in your study? What is the ensemble size? What are the independently observed degrees of freedom in your case?

L146: stability of the *EnKF*?

L150: Add more detail about what exactly you mean by *sampling errors.*

L151: What localization and inflation parameters are you examining?

Sec. 2.3: You either need to embed this information into the previous section or clearly outline at the beginning of the methods section that you are first describing the EnKF in general and then how this general structure relates to your specific setup (with references to sections). Again, a flowchart linking the general structure to your experiments would be helpful.

L154: EnKF or EAKF? If EnKF, then why bother introducing EAKF?

L164: Why did you decide to use lower standard deviations? What is the plausible range?

L178-179: Are you assuming that the friction coefficient and bed topography are uncorrelated?

L184: What radii did you explore?

L186: Ensemble size of 30 to 100, but what steps exactly?

Fig. 3: The font size is too small. This is generally the case for a lot of plots and I will refrain from mentioning it again afterwards. Otherwise, I think this is a great figure supporting the description of your OSSEs.

L221 – *reference model mesh*: Do you mean the mesh used in the reference simulation?

L230-231: This information needs to come earlier.

L234: Can you provide any insight as to why these values lead to the minimum RMSEs?

L238: Why is that expected?

L240: You are using *inflation parameters* in the text but *inflation factor* in Fig.4.

L243-245: For ensemble size 100, the RMSE for friction coefficient does NOT continue to decrease steadily (increase at t=7a). The other two ensemble sizes also show a small increase just after 5 years. Similar peaks are also visible for the bed topography and ice thickness. What is causing this increase in RMSE?

L245: A larger ensemble size (100 vs. 50) actually leads to a larger RMSE after t=15 a for all panels in Fig. 5. Why do you think this is the case? And what does it mean for the design of future experiments?

L248: Why did you choose a localization radius of 4 km when bed topography and ice thickness showed a minimum RMSE for 6 km with a significant increase for smaller radii?

L250: Somewhere you should state explicitly that a localization radius of 4 km and an inflation parameter of 1.12 are your *optimal DA configuration*.

L254-256: You describe the results for the friction coefficient and bed topography, but what about the ice thickness?

L257-262: This is a description of your experimental design and should be in the methods section.

L264: Just to make sure I understood it correctly. The deterministic forecast uses the mean (across all ensemble members) basal friction coefficient and mean bed topography but is a single simulation (compared to running a simulation for all ensemble members and then calculating the ensemble mean).

L268-269: I think it is quite interesting that the deterministic forecast, which is based on the ensemble mean basal conditions, follows the reference simulation relatively closely while most of the individual ensemble member simulations show a much smaller ice volume change. I suspect this is due to non-linearities in the system, but it might be worth having a closer look at this.

L271-273: So assimilating more observations leads initially to a better agreement but increases the difference in ice volume change at the end of the forecast period. This should be addressed in more detail in the discussion.

L273-274: Discussing the increase in the rate of mass loss after 100 a and its implication for sea level rise projections over the next century (compared to 200 yr projections) could be a nice additional takeaway.

Fig. 4: Why are some experiments diverging? Why does the diverging area shift to smaller inflation factors as the ensemble size increases? What is the reason for the sharp increase in RMSE for localization radii smaller than the minimum RMSE? Use friction coefficient, bed elevation, and ice thickness as labels next to the colour bar.

Fig. 5: I suggest using RMSE_C, RMSE_B, and RMSE_H as y-labels or adding friction coefficient, bed elevation, and ice thickness as titles. The description of panel c is missing. The colour coding is somewhat confusing because you are using the same colours as in Fig. 4 but they do not represent the same thing (ensemble size vs. friction coefficient/bed elevation/ice thickness).

Fig. 6: Why did you choose a more or less symmetrical (about y=40 km) friction coefficient but a very asymmetrical bed topography? The units are missing in the colour bar label. You might want to increase the spacing between panels to make it clearer which text corresponds to which panel.

Fig. 7: What causes the sharp grounding line extent towards higher x values at y=10 km in the *no assimilation* panel? Units are missing. Increase spacing between panels.

Fig. 8: Why did you not show the difference in Fig. 6 and 7? What causes the *checkerboard* pattern?

Fig. 9: I suggest adding a legend for the different lines. Change *reference run* to *reference simulation*. Change *forecast simulation* to *deterministic forecast simulation* (all of them are forecast simulations).

Sec. 3.2.: Do not use abbreviations as section titles.

L285: Change *10 to 15 km* to *15 km*. Or did you also test, e.g., 13 km across-track spacing?

L286: I am not sure what you mean by *the performance declines due to suboptimal choices for inflation and localization parameters*. You are examining the effects of these parameters here, so shouldn't you be able to determine the optimal choices? Do you mean the optimal choices are outside your tested parameter ranges? If so, you need to show results supporting this claim.

L287: Start a new paragraph before *For the gridded*.

L288: Add the reference to Fig. 11 to the end of this sentence (currently at the end of the paragraph).

L289: range *of* localization

L292 – *we conducted*: Use present tense throughout the manuscript (e.g., same issue in L403 *presented*).

Fig. 10: In panels (f) and (i), why is the RMSE of localization radius 6 km and inflation factor 1.04 much larger than the surrounding values?

L297-299: Add reference to Table 2.

L301 – *RMSE values continue to decrease until the end*: Add reference to Fig. 12. Panels c, f, and i show an increase in RMSE at the end.

L301-303: Add reference to Table 3.

L304 – *marginal improvements ... after 10 - 15 years*: Add reference to Fig. 13. Again, RMSE actually increases in panels f and i. What do you think causes this increase?

L304-305: This is discussion.

L307: Add reference to Table 2. For RMSE_C, the smallest uncertainty leads to the second-largest RMSE of all tested uncertainties. So DA performance does not necessarily decrease as uncertainty increases!

L308-309: Add reference to corresponding panel in Fig. 12. For the 10 km across-track data, the maximum difference in RMSE_C is 15.43. For the 5 km data, it is only 8.65. Although the maximum difference is much smaller in the 5 km case, you argue that it shows a decrease in performance as uncertainty increases while the larger 10 km difference indicates a consistent performance across different uncertainty levels. I believe your statement is primarily based on the RMSE_B and RMSE_H results, but these details need to be spelled out!

L309-311: Adding to my previous comment, if you compare Track_15km_$\sigma_h$_10_$\sigma_v$_10 to Track_15km_$\sigma_h$_15_$\sigma_v$_10, and Track_15km_$\sigma_h$_20_$\sigma_v$_10, the DA performance increases as uncertainty increases for RMSE_C, RMSE_B, and RMSE_H. This needs to be stated clearly and discussed in detail!

L311: Add a reference to the corresponding panel in Fig. 12. They actually show an increase in RMSE after 15 to 20 years.

L311: Add new paragraph before *With the 1 km gridded*.

L312: Add a reference to Table 3. What about the friction coefficient? For the friction coefficient and Track_15km, the highest uncertainty level has the smallest RMSE and, therefore, the best performance. So DA performance does not necessarily decrease as uncertainty increases!

L313 – *DA performance does not vary significantly across different uncertainty levels*: I don't think I agree with this statement. Again, just looking at RMSE_C, the maximum difference across uncertainty levels at 1 km resolution is 2.84, while it is 10.28 at 10 km and 30.8 at 20 km. So if anything, the performance varies more significantly with coarser grid resolution. Additionally, the RMSE_C for the coarsest grid resolution (20 km) and highest uncertainty level is smaller than all other RMSE_C values at 20 km AND 10 km resolution! As the RMSE_C with the highest uncertainty and the coarsest across-track resolution is also smaller than all other values at this resolution, it seems unlikely that this is just a coincidence. So this really needs to be addressed.

Fig. 12: Use RMSE_C, RMSE_B, and RMSE_H in y-labels. Panel labels in the second and third rows are the same. Actually, even the subplots themselves look the same. The panel labels seem to be just a copy/paste issue in your Python code, not sure about the actual data.

Fig. 13: Same issues as for Fig. 12. What causes the increase in panel f after 20 years? Why is there such a rapid increase in RMSE in, e.g., panel d between 5 and 10 years? Why does this rapid increase get muted for coarser resolutions? A similar pattern occurs in Fig. 12.

L316 – *fast flowing regions*: You haven't mentioned fast-flowing regions before. Are you referring to *areas around the grounding line, where the signal-to-noise ratio of velocity is relatively high*? I'd argue that large differences also occur for y=70-80 km and x=450-640 km, which seems to be a relatively slow-flowing region.

L319 – *fast flowing regions*: Again, what about the region between y=70-80 km and x=450-640 km?

L322: *more* accurate.

L333 – *assumptions on the initial ensemble*: Be more specific.

L337 – *relatively small ensemble size*: Be precise.

L337-338 – *previous studies*: References are missing.

L342 – *a larger ensemble size could provide advantageous*: Or not. Fig. 5e shows a larger RMSE for 100 members than 50 members by t=30 years.

L343: What exactly should these studies investigate to identify the optimal approach?

L345 – *similar to values from earlier studies*: References are missing. What are these values?

L347: 4-120 km is a lot wider range than your 4-8 km. What causes these differences?

L351: ... assimilating more observations, *i.e. more assimilation years*, to estimate ...

L352 – *improves accuracy of model projections*: In general, yes, but the difference in ice volume change at t=200 years is larger in Fig. 9 panel b than panel a (between red and blue line).

L354 – *XX*: Add numbers.

L358 – *in this study that*: Replace with *here.*

L358 – *observations maintains*: Replace with *observations while maintaining*

L361: I suggest restating what OSSE means for readers quickly skimming through the manuscript.

L361: Remove *in this study.*

L361 – *the capabilities of OSSEs*: Replace with *their capabilities*.

L365 – *(Table 2 and 3)*: If you include references here then you should also include them in previous paragraphs of the discussion.

L365-366: As indicated previously, this is not the case for RMSE_C in Table 3.

L369-371: As mentioned above, additional data points can actually have a negative effect.

L373-374: I disagree. The friction coefficient has the largest differences in RMSE across uncertainty levels (for all resolutions in Table 2 and similar for Table 3), so it is actually the most sensitive.

L377: *will* or *should*?

L379-380: Your *future studies should address* sentences are spread out across the entire discussion. I recommend bundling all of them into one single paragraph at the end of the discussion.

L398-399: Again, this is not always the case.

L400: What does *great accuracy* mean? Be precise.

L404 – *Different levels of observational uncertainty*: Do you mean *smaller levels of uncertainty*?

L410: Will you also upload the data files?

I hope the authors find my comments helpful.

Sincerely, Kevin Hank

---

## Author Comment (AC1)

**Estimation of the state and parameters in ice sheet model using an ensemble Kalman filter and Observing System Simulation Experiments**
**– Authors' response (RC1) –**

Youngmin CHOI et al.

May 1, 2025

*General comments*

*The manuscript by Choi et al. presents a data assimilation framework to improve the projection capabilities of ice sheet models. Specifically, the performance of an Ensemble Adjustment Kalman Filter in constraining the model state (ice thickness) and basal conditions (basal friction coefficient and bed topog- raphy) of a 2D plan-view ice model is assessed. Their results indicate that assimilating more observations generally increases the accuracy of model projections, with projections for up to 200 years in close agreement with the reference simulation. The performance of the data assimilation method is sensitive to the observational error as well as the cross-track spacing and grid resolution of surface elevation data.*

*I believe the science behind this study is sound and aligns with the focus of The Cryosphere (TC). However, the presentation of the methodology lacks clarity, at times adding avoidable confusion (e.g., the introduction of both acronyms EnKF and EAKF). This overall issue is addressed in more detail in the specific comments below, but I strongly suggest the addition of a flowchart outlining the methodology (ice sheet model and data assimilation) and experimental design (twin experiment and OSSEs).*

We thank the reviewer for reviewing the manuscript and constructive comments. We will thoroughly address specific comments and aim to clarify all points as clearly as possible. We appreciate the suggestion to include a flowchart outlining the methodology and will add it into the revised manuscript.

*Furthermore, parts of the experimental design are currently placed within the results section and*
*key aspects of the results are not addressed (e.g., why is the RMSE C for Grid_20 km_$\sigma_h$_20_$\sigma_v$_10*
*smaller than for the same grid resolution with smaller uncertainties as well as all 10 km grid reso-*
*lution experiments?). Considering the performance of the data assimilation method is sensitive to*
*the uncertainty in surface elevation observations, I believe also determining the effect of various*
*uncertainties in the velocity data would add further value to the manuscript (perhaps as supple-*
*mentary material). Finally, potential reasons/explanations for model results are often missing, e.g.,*
*why is the range of optimal localization radius (4 - 8 km) a lot smaller than suggested by previous*
*studies (4 - 120 km)? I recommend the authors also take my specific comments listed below into*
*account.*

We agree with the reviewer that the some of the results lack sufficient explanation. We will address
this in the revised manuscript. Furthermore, we will conduct additional experiments with varying
uncertainties in the velocity data and include the new results in the updated version.

*Specific comments*

*L25: This sentence is very similar to the second sentence in the introduction. Instead, consider*
*opening with a sentence about the different DA methods (variational vs. methods leveraging time-*
*varying observations). Then proceed to discuss advantages/disadvantages of each.*

We will revise this sentence.

*L28: Double brackets.*

We will fix this.

*L33: Consider starting a new paragraph before Alternatively.*

We will make this a new separate paragraph as suggested.

*L37 – assimilation period: Readers unfamiliar with DA might not know what exactly you refer to*
*here. It becomes a lot clearer later on, but it would be nice to have a brief definition here (similar*
*for other DA-specific terms, e.g., data denial experiments in L61).*

We will include brief definitions of several DA terms for clarity.

*L40: Move further up to the rest of the discussion on variational methods.*

We will consider rewriting this paragraph to improve clarity.

*L42: Consider introducing the term ensemble DA in general before describing the specific EnKF (e.g., ensemble DA vs. variational methods).*

We will add introductory sentences to clarify the ensemble DA.

*L54: Why are ensemble DA methods less commonly used in ice sheet modelling?*

Ensemble DA methods are less commonly used in ice sheet modeling primarily due to historical limitations in observational data and computational cost. Ensemble approaches rely on consistent, time-varying observations with well-characterized uncertainties. However, surface observations for ice sheets have often been sparse, noisy, or temporally inconsistent, which are less suitable for ensemble DA. The ice sheet modeling community has traditionally relied on adjoint-based (variational) inversion methods for parameter estimation using time-invariant mosaics or composites data (e.g., multi-year averaged surface velocity fields). Additionally, ensemble methods typically require multiple forward model runs, making them more computationally demanding than (static) variational approaches. These challenges led to the relatively limited adoption of ensemble-based DA in ice sheet modeling. We will revise this section in the manuscript accordingly for clarity.

*L63 – (OSSEs)(OSSEs, ...).*

We will fix this.

*L65 – appropriate observation error distribution: How do you determine if the distribution is appropriate or not?*

We meant a prescribed observation error distribution representative of real measurement uncertainties. We will revise this sentence to clarify our intent.

*L71: Although it is addressed in more detail in the next sentence, I believe adding (ice thickness) just after model state would add clarity.*

We will add that.

*L75 – estimated state and parameters: For consistency, I recommend using estimated model state and parameters throughout the manuscript.*

We will use consistent terminology throughout the manuscript.

*L75 – true reference values: At this point in the manuscript, it is not clear what the true reference values are and how you obtain them.*

We will consider rephrasing this sentence for clarity.

*L83-84: Remove this sentence and add the reference to the description of the specific sections in*
*the text above.*

We will remove this sentence.

*L86-88: Repetition of the text just before the Methods section.*

We will revise these sentences.

*L101: I am not familiar with this specific method, but a standard deviation of 500 m seems quite*
*large considering the bed varies only between zb,deep = -720 m and ∼500 m (really difficult to see*
*in Fig. 1a)*

The midpoint displacement method generates a 2D surface by iteratively subdividing a grid, assign-
ing random heights to corner points, and interpolating midpoints with added random displacement.
The magnitude of the displacement is scaled by a standard deviation that decreases with each iter-
ation as $2^{0.5H}$, where $H$ is the roughness factor, set to 0.7 in this study. While the initial standard
deviation of 500 m may seem large relative to the vertical range of the bed topography, it is used
as a starting point in the midpoint displacement algorithm and is progressively reduced at each
iteration based on the roughness factor. This results in a realistic, spatially correlated roughness
pattern with limited high-amplitude variations. Additionally, the current value of $z_{b,deep} = 720m$
represents the base shape of the bed before roughness is added and the final bed elevation reaches
depths of approximately -1,500 m. We will add these details to the text and revise the Fig. 1a as
suggested below.

*Fig. 1: I believe using two separate 2D plots instead of the 3D plot would make the identification*
*of certain details and interpretation of the plot a lot easier. As you are already showing the bed*
*topography in Fig. 2, I recommend combining Fig. 1 and 2 into a single plot with panels a: ice*
*surface elevation, b: bed topography, and c: ice velocity. Note that the rainbow colour scheme is*
*not in line with the journal guidelines. You can check all of your plots with the colour blindness*
*simulator (https://www.color-blindness.com/coblis-color-blindness-simulator/). The fonts in panel*
*b are too small and I recommend using a different colour for your contour lines.*

We will revise the figure to address all of the reviewer's suggestions.

*Fig. 2: Why are you using such an asymmetrical (about y=40 km) bed topography compared to*
*the commonly used symmetrical approach in idealized studies? The y label in Fig. 2 indicates the*

*domain ranges from 20 to 100 km, whereas in Fig. 1 it is 0 to 80 km. You also might want to remove the white margins at the top and bottom. How can the bed elevation be -1500 m when Eq. 1 limits the bed to $z_{b,deep}$ = -720 m?*

Applying the midpoint displacement method results in an asymmetrical bed topography, which may better reflect realistic subglacial features, although we use an idealized twin experiment in this study. We will include this explanation and revise the figure as suggested.

*L103: Consider adding an additional panel (d) showing the triangular mesh to the new Fig. 1. Are you using adaptive mesh refinement, e.g., following the grounding line?*

We use an adaptive mesh based on ice velocity and will include a new figure for the mesh in the revised manuscript.

*L104: I suggest adding another panel (e) for the basal friction coefficient to Fig. 1 or at least refer to Fig. 6a here.*

We will refer to Fig. 6a to avoid repetition.

*Eq. 5: In case you are working in LaTeX, I recommend using left( and right) to get brackets of the correct size.*

We will fix the bracket.

*Eq. 6: Same as for Eq. 5*

We will fix the bracket.

*L114: Is C in Eq. 7 different from the one described in Eq. 4? If not, then remove C is a friction coefficient.*

$C$ is the friction coefficient, and $C\_x$ and $C\_y$ are the x and y components of $C$, respectively. We will add this to the text.

*L117: Remove equal.*

We will remove it.

*L124: Do you consider a melt rate of 200 m/yr realistic given that maximum present-day melt rates are around 100 m/yr?*

We set the melt rate to 200 m/yr at a depth of 800 m, which results in an actual melt rate of approximately 170 m/yr beneath the ice shelf. We agree that this melt rate exceeds the maximum observed present-day basal melt rates. However, in this study, we chose this value to create a strong dynamic response in the model over a 200-year forecast period, ensuring that the effects of data assimilation could be clearly evaluated. The elevated melt rate is not meant to represent a realistic present-day climate, but rather to serve as a diagnostic tool in the context of a twin experiment. We will clarify this in the revised manuscript.

*L125-127: This belongs into results.*

This describes the process of creating the reference run for the twin experiment rather than presenting model results. We will revise the text to clarify it.

*L129: Sec. 2.3 and 2.4 are referenced before 2.2.*

We will delete this sentence here.

*L134: modified version of the Ensemble Kalman Filter? As I mentioned above, using EnKF and EAKF is confusing, especially since EAKF is introduced but only used within this paragraph.*

We will clarify the use of data assimilation terminology throughout the manuscript.

*L136-143: This is where I think a flowchart would really help the reader to follow the details of your method. Ideally, the flowchart should outline the details of the EAKF and how it relates to your specific study. For example: How do ensemble members differ? What exactly is your model forecasting? How is the observation window specified? Which ice sheet variables are considered in the state vector? What are the state variables?*

We will include a detailed flowchart to clarify the methodology.

*L142: How would adding extra variables, like surface velocity, to the state vector affect your results?*

Surface velocity is an observation we assimilate in our study, but it is not part of the state vector. We will clarify this in the text.

*L143-144: EnKFs or EAKFs? Does this challenge arise in your study? What is the ensemble size? What are the independently observed degrees of freedom in your case?*

Here, we are explaining the general case for the ensemble Kalman filter. As mentioned above, we will revise this paragraph to improve clarity and better distinguish the general description from our specific implementation.

*L146: stability of the EnKF?*

Yes, we will change it.

*L150: Add more detail about what exactly you mean by sampling errors.*

In the revised manuscript, we will clarify that sampling errors in ensemble based data assimilation arise due to the limited number of ensemble members used to approximate the forecast error covariance.

*L151: What localization and inflation parameters are you examining?*

We will add details about the localization and inflation methods here, including the associated tuning parameters for each.

*Sec. 2.3: You either need to embed this information into the previous section or clearly outline at the beginning of the methods section that you are first describing the EnKF in general and then how this general structure relates to your specific setup (with references to sections). Again, a flowchart linking the general structure to your experiments would be helpful.*

We will clearly outline the description of the methods at the beginning of the Methods section.

*L154: EnKF or EAKF? If EnKF, then why bother introducing EAKF?*

EnKF is the general term, and EAKF is the specific approach we use in this study. We will clarify this in the text.

*L164: Why did you decide to use lower standard deviations? What is the plausible range?*

We selected lower standard deviation values (5 m for surface elevation and 10 m/yr for velocity) to provide a simple and conservative baseline for the twin experiment. While these values are lower than those used in Gillet-Chaulet (2020), they are still within the plausible observational uncertainty ranges reported in recent literature. For example, Dai and Howat (2017) report vertical elevation uncertainties below 5 m in well-constrained regions, and Mouginot et al. (2017) report horizontal velocity uncertainties ranging from 5–20 m/yr depending on the region. We chose values at the lower end of these ranges to isolate the performance of the DA framework under favorable conditions, and we explore sensitivity to larger uncertainties in the OSSEs presented in Section 3.2. We will clarify this in the revised manuscript.

*L178-179: Are you assuming that the friction coefficient and bed topography are uncorrelated?*

They are correlated, as both parameters are sensitive to common observations. We will include this explanation in the revised manuscript.

*L184: What radii did you explore?*

The radii explored in the experiment ranged from 2 km to 20 km. We will add this information to the text.

*L186: Ensemble size of 30 to 100, but what steps exactly?*

We tested ensemble sizes of 30, 50, and 100, and will specify this information clearly in the text.

*Fig. 3: The font size is too small. This is generally the case for a lot of plots and I will refrain from mentioning it again afterwards. Otherwise, I think this is a great figure supporting the description of your OSSEs.*

We will increase the font size.

*L221 – reference model mesh: Do you mean the mesh used in the reference simulation?*

Yes, we will clarify this in the text.

*L230-231: This information needs to come earlier.*

We will move this information to the Method section.

*L234: Can you provide any insight as to why these values lead to the minimum RMSEs?*

The localization radius is determined through a set of sensitivity experiments and is based on the expected spatial correlation length scale of the parameters, which may depend on the size of flow features or stress balance regimes. We will add further discussion on this point in the revised discussion section.

The smaller ensembles tend to underestimate the ensemble spread due to sampling errors, which can lead to filter divergence. To compensate for this underestimation, higher inflation factors are often required to maintain sufficient ensemble variance. As the ensemble size increases, the sampling error is reduced, leading to more accurate estimation of error covariances and therefore requiring less inflation. We will clarify this explanation in the revised manuscript.

*L240: You are using inflation parameters in the text but inflation factor in Fig.4.*

We will clarify the term for consistency.

*L243-245: For ensemble size 100, the RMSE for friction coefficient does NOT continue to decrease steadily (increase at t=7a). The other two ensemble sizes also show a small increase just after 5 years. Similar peaks are also visible for the bed topography and ice thickness. What is causing this increase in RMSE?*

We examined the small increase in RMSE in early assimilation years and found that it is likely due to a temporary mismatch between the model forecast and the observations during this period, potentially caused by transient model dynamics or nonlinearities in the response to assimilated observations. As the assimilation continues, the filter gradually corrects these discrepancies, which leads to a subsequent reduction in RMSE. These fluctuations are not uncommon in ensemble data assimilation systems, especially in complex, nonlinear models where localized error growth can temporarily degrade performance. We will discuss this behavior in the revised manuscript.

*L245: A larger ensemble size (100 vs. 50) actually leads to a larger RMSE after t=15 a for all panels in Fig. 5. Why do you think this is the case? And what does it mean for the design of future experiments?*

While larger ensemble sizes generally improve the accuracy of error covariance estimates, they can also increase the sensitivity of the filter to model errors or sampling noise if not properly tuned. In our experiments, it is possible that the inflation and localization parameters used for the 100-member ensemble were not optimal for later assimilation periods, leading to slightly degraded performance after year 15. This suggests that filter performance does not necessarily scale linearly with ensemble size and that tuning DA parameters for each ensemble size is critical. It also emphasizes the importance of adaptive inflation/localization techniques or diagnostics for dynamically adjusting filter settings. We will revise the manuscript to reflect this finding and its implications for future ensemble DA experiment design.

*L248: Why did you choose a localization radius of 4 km when bed topography and ice thickness showed a minimum RMSE for 6 km with a significant increase for smaller radii?*

The minimum RMSE for bed topography and ice thickness occurs at a localization radius of 6 km, but the RMSEs for both parameters at 4 km are also low. We chose 4 km for illustrative purposes and will clarify this in the text.

*L250: Somewhere you should state explicitly that a localization radius of 4 km and an inflation parameter of 1.12 are your optimal DA configuration.*

The optimal inflation factor and localization radius depend on the parameter being estimated. We chose 4 km and 1.12 for illustrative purposes and will clarify this in the text.

*L254-256: You describe the results for the friction coefficient and bed topography, but what about the ice thickness?*

The pattern in the ice thickness results is very similar to that of bed topography. In our model setup, surface elevation is defined as the sum of ice thickness and bed topography (surface = thickness + bed). Therefore, as surface observations are assimilated, improvements in bed estimates are reflected in the estimated thickness field. We will clarify this relationship and summarize the results for ice thickness more explicitly in the revised manuscript.

*L257-262: This is a description of your experimental design and should be in the methods section.*

We will move this to the Methods section.

*L264: Just to make sure I understood it correctly. The deterministic forecast uses the mean (across all ensemble members) basal friction coefficient and mean bed topography but is a single simulation (compared to running a simulation for all ensemble members and then calculating the ensemble mean).*

Yes. we will clarify it in the text.

*L268-269: I think it is quite interesting that the deterministic forecast, which is based on the ensemble mean basal conditions, follows the reference simulation relatively closely while most of the individual ensemble member simulations show a much smaller ice volume change. I suspect this is due to non-linearities in the system, but it might be worth having a closer look at this.*

We agree with the reviewer's point. The observed behavior likely results from the nonlinearities of the model. While each ensemble member represents a physically plausible realization of the basal parameters, small deviations from the true field can lead to large differences in modeled ice volume due to non-linear feedbacks. However, the deterministic forecast, initialized with the ensemble mean of the basal fields, appears to capture the overall structure of the true conditions more effectively, reducing local extremes and yielding results that are more closer to the reference simulation. We will add this discussion to the revised manuscript.

*L271-273: So assimilating more observations leads initially to a better agreement but increases the difference in ice volume change at the end of the forecast period. This should be addressed in more detail in the discussion.*

Assimilating more observations leads to better agreement throughout the forecast period, including at its end. We will include specific values for ice volume loss to support this comparison in the revised manuscript.

*L273-274: Discussing the increase in the rate of mass loss after 100 a and its implication for sea level rise projections over the next century (compared to 200 yr projections) could be a nice additional takeaway.*

We will add this to the discussion section.

*Fig. 4: Why are some experiments diverging? Why does the diverging area shift to smaller inflation factors as the ensemble size increases? What is the reason for the sharp increase in RMSE for localization radii smaller than the minimum RMSE? Use friction coefficient, bed elevation, and ice thickness as labels next to the colour bar.*

When the localization radius is too small, it overly restricts the influence of observations on the state update. This can lead to underestimation of error covariances and result in filter divergence. In our experiments, this is evident when the localization radius falls below the specific threshold of each variable (e.g., 4 km for friction and 6 km for bed topography). We will include this explanation in the revised manuscript. We will also revise the figure as suggested.

*Fig. 5: I suggest using RMSE_C, RMSE_B, and RMSE_H as y-labels or adding friction coefficient, bed elevation, and ice thickness as titles. The description of panel c is missing. The colour coding is somewhat confusing because you are using the same colours as in Fig. 4 but they do not represent the same thing (ensemble size vs. friction coefficient/bed elevation/ice thickness).*

We will revise the figure as suggested.

*Fig. 6: Why did you choose a more or less symmetrical (about y=40 km) friction coefficient but a very asymmetrical bed topography? The units are missing in the colour bar label. You might want to increase the spacing between panels to make it clearer which text corresponds to which panel.*

We will revise the figure as suggested.

*Fig. 7: What causes the sharp grounding line extent towards higher x values at y=10 km in the no*
*assimilation panel? Units are missing. Increase spacing between panels.*

The asymmetrical grounding line position is caused by the asymmetrical bed topography. We will
revise the figure as suggested.

*Fig. 8: Why did you not show the difference in Fig. 6 and 7? What causes the checkerboard*
*pattern?*

We chose to show the difference in ice thickness in Fig. 8 because changes in thickness are difficult
to detect visually from the similar figure as Fig. 6 and 7. The artifacts observed in the ice thickness
are the result of the conditional random fields generated using the Kriging method, which can
produce "bull's eye" patterns commonly observed between observation points. We will clarify this
in the revised manuscript.

*Fig. 9: I suggest adding a legend for the different lines. Change reference run to reference sim-*
*ulation. Change forecast simulation to deterministic forecast simulation (all of them are forecast*
*simulations).*

We will revise the figure as suggested.

*Sec. 3.2: Do not use abbreviations as section titles.*

We will change the title.

*L285: Change 10 to 15 km to 15 km. Or did you also test, e.g., 13 km across-track spacing?*

We will revise it as suggested.

*L286: I am not sure what you mean by the performance declines due to suboptimal choices for*
*inflation and localization parameters. You are examining the effects of these parameters here, so*
*shouldn't you be able to determine the optimal choices? Do you mean the optimal choices are*
*outside your tested parameter ranges? If so, you need to show results supporting this claim.*

We will remove the phrase "suboptimal choices for parameters" and rephrase this paragraph.

We will revise it as suggested.

*L288: Add the reference to Fig. 11 to the end of this sentence (currently at the end of the para-graph).*

We will revise it as suggested.

We will revise it.

*L292: we conducted: Use present tense throughout the manuscript (e.g., same issue in L403 presented).*

We will revise the manuscript to consistently use the present tense.

*Fig. 10: In panels (f) and (i), why is the RMSE of localization radius 6 km and inflation factor 1.04 much larger than the surrounding values?*

The RMSE values are calculated over the entire domain, and localized errors can increase the overall RMSE. We will include this discussion in the revised manuscript.

We will add the reference.

*L301: RMSE values continue to decrease until the end: Add reference to Fig. 12. Panels c, f, and i show an increase in RMSE at the end.*

We will add the reference.

We will add the reference.

*L304: marginal improvements ... after 10–15 years: Add reference to Fig. 13. Again, RMSE actually increases in panels f and i. What do you think causes this increase?*

We will add the reference. We examined the small increase in RMSE near the end of the assimilation period and found that it is likely due to a temporary mismatch between the model forecast and the observations, potentially caused by transient model dynamics or nonlinearities in the response to assimilated observations. We will discuss this behavior in the revised manuscript.

*L304-305: This is discussion.*

We will move this to the Discussion section.

*L307: Add reference to Table 2. For RMSE_C, the smallest uncertainty leads to the second-largest RMSE of all tested uncertainties. So DA performance does not necessarily decrease as uncertainty increases!*

We agree with the reviewer that this paragraph is not well presented. We will revise it to clearly reference Table 2 or 3 and Fig. 12 or 13 separately, in order to distinguish between the RMSE values at the end of the assimilation period and the changes in RMSE during the assimilation period.

*L308-309: Add reference to corresponding panel in Fig. 12. For the 10 km across-track data, the maximum difference in RMSE_C is 15.43. For the 5 km data, it is only 8.65. Although the maximum difference is much smaller in the 5 km case, you argue that it shows a decrease in performance as uncertainty increases while the larger 10 km difference indicates a consistent performance across different uncertainty levels. I believe your statement is primarily based on the RMSE_B and RMSE_H results, but these details need to be spelled out!*

We are referring here to changes in RMSE values during the assimilation period. We will clarify this in the revised manuscript.

*L309-311: Adding to my previous comment, if you compare $Track\_15km\_\sigma_h\_10\_\sigma_v\_10$ to $Track\_15km\_\sigma_h\_15\_\sigma_v\_10$, and $Track\_15km\_\sigma_h\_20\_\sigma_v\_10$, the DA performance increases as uncertainty increases for RMSE_C, RMSE_B, and RMSE_H. This needs to be stated clearly and discussed in detail!*

Again, we are referring here to changes in RMSE values during the assimilation period. We will rephrase this paragraph to clearly separate the results shown in Table 2 and Figure 12.

*L311: Add a reference to the corresponding panel in Fig. 12. They actually show an increase in RMSE after 15 to 20 years.*

We will add the reference. Again, we will add discussion on this increase in RMSE.

*L311: Add new paragraph before "With the 1 km gridded".*

As suggested, we will separate the discussion of the 1 km results into a new paragraph.

*L312: Add a reference to Table 3. What about the friction coefficient? For the friction coefficient and Track_15km, the highest uncertainty level has the smallest RMSE and, therefore, the best performance. So DA performance does not necessarily decrease as uncertainty increases!*

We will reference Table 3 and explain that the smallest uncertainty does not consistently produce the lowest RMSE, reflecting possible nonlinearities or localized sensitivities.

*L313: DA performance does not vary significantly across different uncertainty levels: I don't think I agree with this statement. Again, just looking at RMSE_C, the maximum difference across uncertainty levels at 1 km resolution is 2.84, while it is 10.28 at 10 km and 30.8 at 20 km. So if anything, the performance varies more significantly with coarser grid resolution. Additionally, the RMSE_C for the coarsest grid resolution (20 km) and highest uncertainty level is smaller than all other RMSE_C values at 20 km AND 10 km resolution! As the RMSE_C with the highest uncertainty and the coarsest across-track resolution is also smaller than all other values at this resolution, it seems unlikely that this is just a coincidence. So this really needs to be addressed.*

We intended to refer to DA performance in estimating bed topography and thickness in this paragraph. We will rephrase it to incorporate all suggestions raised by the reviewer.

*Fig. 12: Use RMSE_C, RMSE_B, and RMSE_H in y-labels. Panel labels in the second and third rows are the same. Actually, even the subplots themselves look the same. The panel labels seem to be just a copy/paste issue in your Python code, not sure about the actual data.*

We will revise the figure as suggested. The pattens for RMSE_B and RMSE_H over assimilation time is very similar to each other since surface elevation is defined as the sum of ice thickness and bed topography (surface = thickness + bed). The figures in the second and third lows look very similar but not the same figures (note that values in y axis).

*Fig. 13: Same issues as for Fig. 12. What causes the increase in panel f after 20 years? Why is there such a rapid increase in RMSE in, e.g., panel d between 5 and 10 years? Why does this rapid increase get muted for coarser resolutions? A similar pattern occurs in Fig. 12.*

As mentioned above for other figures (e.g., Fig. 5), it is likely due to a temporary mismatch between the model forecast and the observations during this period, potentially caused by transient model dynamics or nonlinearities in the response to assimilated observations. Localized error growth can temporarily degrade performance. We will discuss this point in the revised manuscript.

*L316: fast flowing regions: You haven't mentioned fast-flowing regions before. Are you referring to areas around the grounding line, where the signal-to-noise ratio of velocity is relatively high? I'd argue that large differences also occur for y=70-80 km and x=450-640 km, which seems to be a relatively slow-flowing region.*

We will clarify the fast flowing regions in the revised manuscript.

*L319: fast flowing regions: Again, what about the region between y=70-80 km and x=450-640 km?*

We will clarify the fast and slow flowing regions in the revised manuscript.

*L322: "more" accurate.*

We will revise this as suggested.

*L333: assumptions on the initial ensemble: Be more specific.*

We meant how the initial ensemble is generated. We will revise this.

*L337: relatively small ensemble size: Be precise.*

We will specify the ensemble size.

*L337-338: previous studies: References are missing.*

We will add the references.

*L342: a larger ensemble size could provide advantageous: Or not. Fig. 5e shows a larger RMSE for 100 members than 50 members by t=30 years.*

We agree that increasing ensemble size does not always guarantee improved DA performance. We will revise the discussion to acknowledge that larger ensemble sizes can improve performance in general but may also introduce challenges that must be carefully managed, particularly in long assimilation periods or highly nonlinear systems.

*L343: What exactly should these studies investigate to identify the optimal approach?*

We will remove this sentence, as a rephrased paragraph discussing future studies will be included, as suggested by the reviewer.

L345: similar to values from earlier studies: References are missing. What are these values?

We will add the references.

L347: 4-120 km is a lot wider range than your 4-8 km. What causes these differences?

The differences in the optimal localization radius likely comes from the differences in model configuration, dimensionality, and spatial resolution. Our study uses a 2D unstructured mesh with relatively fine spatial resolution, whereas previous studies using flowline models (1D) with coarser grids may require broader localization to account for longer correlation length scales. We will clarify this in the revised manuscript.

L351: ... assimilating more observations, i.e. more assimilation years, to estimate ...

We will revised it as suggested.

L352: improves accuracy of model projections: In general, yes, but the difference in ice volume change at t=200 years is larger in Fig. 9 panel b than panel a (between red and blue line).

We specified "up to 100 years". Up to 100 years, the determinist ice volume loss forecast in Fig. 9b shows better agreement with the reference simulation than in Fig. 9a.

We will add missing values here.

We will revise this as suggested.

L358: observations maintains: Replace with observations while maintaining

We will revise this sentence as suggested.

L361: I suggest restating what OSSE means for readers quickly skimming through the manuscript.

We will revise this as suggested.

*L361: Remove "in this study".*

We will revise this as suggested.

*L361: the capabilities of OSSEs: Replace with their capabilities.*

We will revise this as suggested.

*L365: (Table 2 and 3): If you include references here then you should also include them in previous paragraphs of the discussion.*

We will remove these references here.

*L365-366: As indicated previously, this is not the case for RMSE_C in Table 3.*

We will address this point as the previous mentioned in our response to the comments on Table 3.

*L369-371: As mentioned above, additional data points can actually have a negative effect.*

We will address this point and related it to the temporal decline in DA performance.

*L373-374: I disagree. The friction coefficient has the largest differences in RMSE across uncertainty levels (for all resolutions in Table 2 and similar for Table 3), so it is actually the most sensitive.*

We intended to compare DA performance in estimating bed topography (and ice thickness) versus the friction coefficient. We will add the relevant results and revise the discussion in the revised manuscript.

*L377: will or should?*

We will use "should" instead of "will".

*L379-380: Consolidate all future study suggestions into one paragraph.*

We will combine this paragraph with the next one into a single paragraph.

*L398-399: Your future studies should address sentences are spread out across the entire discussion. I recommend bundling all of them into one single paragraph at the end of the discussion.*

We will consolidate the mentions of future work into a single paragraph at the end of the discussion section to improve clarity.

*L398-399: Again, this is not always the case.*

Except for the temporary decreases in DA performance observed during the assimilation period, this statement generally holds true. We will clarify this point in the revised manuscript.

*L400: What does great accuracy mean? Be precise.*

We agree with the reviewer and will replace "great accuracy" with a quantitative assessment.

*L404: Different levels of observational uncertainty: Do you mean smaller levels of uncertainty?*

It is not necessarily smaller uncertainty levels; it also depends on the resolution or the track spacing of the data. We will clarify this in the revised manuscript.

*L410: Will you also upload the data files?*

Yes, we will upload the data files to the repository.

---

## Author Comment (AC2)

**Estimation of the state and parameters in ice sheet model using an ensemble Kalman filter and Observing System Simulation Experiments**
**– Authors' response (RC2) –**

Youngmin CHOI et al.

May 1, 2025

*This is a review of "Estimation of the state and parameters in ice sheet model using an ensemble Kalman filter and Observing System Simulation Experiments" by Choi et al., submitted for publication to The Cryosphere. This manuscript describes the use of an ensemble-based data assimilation system, the Ensemble Kalman Filter (EnKF), to assimilate data into a 2D large-scale ice sheet models, for the purpose of better estimating parameter values and state variables during the historical period. It follows on other studies that have explored similar methods for 1D ice sheet models, and makes the crucial step of applying such methods to a model widely used for projections. This study also adds a novel "Observing System Simulation Experiment" in which different potential observing system configurations (resolution, track spacing, observational accuracy) are tested to determine their ability to improve accuracy in estimated parameters and state.*

*Overall, I think this is a pretty straightforward study using well-known tools in a new way with ice sheet models, advancing the state of the art in our field. My main suggestions are to further explore certain DA and modeling choices that are unexamined in the current version of the manuscript. I have detailed these suggestions and more minor ones below.*

We thank the reviewer for reviewing the manuscript and constructive comments. We will revise the manuscript to include additional justification for key data assimilation and modeling choices, clarify methodological decisions, and expand the discussion on the implications and limitations of our approach. We address each specific comment in detail below and aim to improve the clarity.

*1. The manuscript briefly describes what the EnKF is, and then indicates that the EAKF version is chosen for this study. There are multiple different flavors of the EnKF available in DART, so it*

*is unclear why EAKF is chosen and whether the results would be any different if another filter was chosen. My suggestion is to describe in some more detail what is done in an EnKF and how the EAKF is different from the standard EnKF. Additionally, either some justification for why the EAKF was chosen and some level of justification for why that is the preferred approach when others are available.*

Thank you for the suggestion; this was also raised by another reviewer. We will add more details about the EnKF and EAKF, and clearly describe the distinctions between the two approaches.

*2. One thing that is unclear from your study design is the relative importance of assimilation window (e.g. 5 vs 15 vs 30 years) as compared to number of assimilation cycles. You don't change the frequency of observations, which may be sensible given than annual observations are reasonable for current observing platforms. However, it is then hard to understand as a reader whether there is something fundamental about having 20-30 years of observations related to the time scales of ice sheet response to adjustments, or whether it is having 20-30 assimilation cycles to improve. If the observations were more frequent (e.g. an IceSAT2-like 90 days) would it takes less time for the EnKF to improve to the level that you show here?*

This is a great point. In this study, we aimed to estimate two constant-in-time parameter fields and the model state on an annual basis. Given the timescales associated with the model state and parameters, as well as the capabilities of current observational platforms, we chose to use annual observations for simplicity.

We agree that the distinction between the length of the assimilation window and the number of assimilation cycles needs further investigation. To address this, we will conduct additional experiments to explore the relative impact of the number of assimilation cycles versus the time period over which they are applied. We will include these results and a discussion in the revised manuscript.

*3. A big difference between your perfect model design and a real scenario where DA might be applied is that only two constant-in-time parameter fields are unknown. In reality, (e.g.) ice viscosity and climate forcing are also likely to be poorly known (though at least climate forcing is directly observable), and climate forcing (and basal friction) may vary in time. Two possibilities that would be helpful to run some experiments to assess are:*

*(a) if you are mistaken about the values of other parameters, but still only estimate basal friction and topography, will the estimate of basal friction compensate for these other errors (particularly for ice viscosity which trades off quite directly with basal friction in a depth-integrated model) - there is some evidence of such compensation already happening in your estimates, see below*

*(b) could this DART-ISSM configuration be used to estimate multiple parameters at once (I don't see a reason why not, but the performance may not be the same as what is found for the single parameter estimation experiments explored currently).*

*I get that the design of these experiments are meant to mimic and compare directly to Gillet-Chaulet*
*2020, but it would be useful to also push beyond their design to get closer to a realistic case where*
*DA might be used.*

Thank you for this insightful comment. We agree that in real-world applications, other parameters such as ice viscosity and climate forcing are also poorly constrained and may vary in space and time. In this study, we intentionally limited the number of unknowns to two constant-in-time parameter fields (basal friction and bed topography) to enable direct comparison with Gillet-Chaulet et al. (2020) and to isolate the behavior of the EnKF framework in a controlled setting.

Regarding (a), we recognize that compensation effects between parameters (e.g., between basal friction and ice viscosity) may arise if other sources of uncertainty are not properly accounted for. We will add a discussion of this limitation and clarify that our experimental design assumes perfect knowledge of other parameters such as ice viscosity, which is not realistic. We also plan to explore the effect of misspecified background parameters (e.g., slightly incorrect viscosity fields) in future sensitivity experiments to assess the potential for parameter compensation.

Regarding (b), we believe that the DART–ISSM framework is capable of estimating multiple parameters simultaneously, and extending the current setup to include more parameters (e.g., ice viscosity, time-varying forcing) is a valuable direction for future work. We will clarify this in the revised manuscript and note that this study provides a foundational step toward that goal by demonstrating the feasibility and utility of ensemble DA with a simplified setup.

*4. At more than one point it is suggested that using variational methods is more computationally intensive that ensemble-based DA. However, there is no real direct proof of this as you don't perform a direct comparison and to my knowledge this has not been done in the published literature. Given that ISSM has a variational DA option already implemented, it could be valuable to compare EnKF with ISSM to EnKF with ISSM in terms of core-hours for a simple standardized run. Short of that, it would be useful to have a sense for the DART overhead? If it is negligible then I would expect ensemble-based DA to have n times the computational expense of a conventional ISSM run where n is the number of ensemble members. Additionally, giving a sense for how this ensemble-based approach has (or can be) parallelized would be useful. In theory, ensemble DA is highly amenable to parallelization, but this depends on how covariance matrices are constructed and how shared memory parallelism is handled. More details on all the computational aspects of this new method would be very useful to include.*

Thank you for the great suggestion. We agree that a computational comparison between the two data assimilation methods would be valuable. However, due to the fundamental differences in the core computational processes of variational and ensemble based approaches, a direct comparison of their computational costs is challenging and may be beyond the scope of this study.

Variational approaches using automatic differentiation (AD) tend to have higher memory demand, while ensemble DA methods primarily increase computational cost through the need to run multiple forward simulations. Additionally, the two approaches are implemented using different tools within ISSM – variational DA is built using the AD tool, while ensemble DA is integrated via DART – which further complicates direct comparison. We will include this limitation in the revised manuscript and mention the need for future work to systematically assess the computational trade-offs between these two methods.

It is also worth noting that DART supports parallel computing. We will also include this in the revised manuscript.

*L16: less numerical model re-development*

We will revise it as suggested.

*L26: use a form of variational*

We will revise it as suggested.

*L28: realy on observational at a single time to*

We will revise it as suggested.

*L30: it often introduces nonphysical artifacts into the*

We will revise it as suggested.

*L35: The use of computational techniques such as automatic differentiation in ice sheet models*

We will revise it as suggested.

*L69: to my knowledge this is the first ice sheet modeling paper to apply OSSE, so I think you can be more direct about this sentence*

We will revise this sentence as suggested.

*L86: an ensemble...for ice sheet model initialization*

We will revise it as suggested.

*L88: on model initialization*

We will revise it as suggested.

*L92: simulation of ice sheets*

We will revise it as suggested.

*L100: explain what the random midpoint displacement method is*

We will add the details with a reference.

*L138: model simulations*

We will revise it as suggested.

*L143: I am confused here because you don't include velocity in the state vector, but later you say*
*it is part of what is assimilated?*

Velocity is an observation being assimilated, not the part of the state vector. We will clarify this in
the text.

*L184: does localization as implemented preserve covariance between different variables at the*
*same location in space or does it simply localize along the diagonal of the covariance matrix?*
*For example, there should be strong correlation between the ice thickness estimate and the bed*
*topography estimate, and so you would be losing a significant amount of your ability to assimilate*
*if covariances between these two variables at the same location in space where zeroed out by*
*localization.*

The localization is applied to the full ensemble covariance matrix, not just along the diagonal.
Therefore, cross-variable correlations at a given location are preserved. We will clarify this point
in the revised manuscript.

*L205: what would happen if you had no velocity observations? How much of the performance is*
*due to velocity observations vs thickness?*

In our current experimental design, we include velocity observations based on the assumption that
high quality annual velocity data are available for most glacier regions. These velocity observa-
tions provide strong constraints on basal friction, particularly in fast-flowing regions. If velocity
observations were removed, we expect the performance of the data assimilation – especially the estimation of the friction coefficient – to degrade, as surface elevation alone provides weaker sensitivity to basal friction. However, the estimation of bed topography and ice thickness may still benefit from surface elevation observations.

We acknowledge that this trade-off between observation types is important for real-world applications. As similar points are raised by another reviewer, we will conduct additional experiments with varying uncertainties in the velocity data and include the new results in the updated version. Additionally, we will emphasize the need for future sensitivity studies to explicitly isolate and quantify the relative contributions of velocity and elevation observations to overall DA performance.

*L243: I think this should refer to Fig. 5*

Yes. We will change it to Fig. 5.

*L259: mean to initialization the deterministics...full ensemble to initialization the ensemble*

We will revise it as suggested.

*Fig 4. In the caption you mention that highly localized experiments diverge. It would be helpful to speak to why these experiments diverge in the main text.*

We will revise the main text to clarify that when the localization radius is too small, it overly restricts the influence of observations on the state update. This can lead to underestimation of error covariances and result in filter divergence. In our experiments, this is evident when the localization radius falls below the specific threshold of each variable (e.g., 4 km for friction and 6 km for bed topography). We will include this explanation in the revised manuscript.

*Fig. 7: There are artifacts in the bed topography and ice thickness estimates that correspond to the basal friction estimate. Can you speak to this? Is it related to how the localization is performed?*

The artifacts in the bed topography and ice thickness are the result of the conditional random fields generated using the Kriging method, which can produce "bull's eye" patterns commonly observed between observation points. We will clarify this in the revised manuscript.

*L276: can you quantify this change in spread? by eye it doesn't seem to change much between 20 and 30 years of assimilation*

We will add the values for each spread in the revised manuscript.

*Figure 9: can you add a legend and plot the ensemble mean as well?*

We will add a legend and include the ensemble mean in the plot.

*L290: It would help to discuss what this sentence means in practice. Is prediction accuracy degraded for this case? Or can you achieve similar results with different localization and inflation parameters?*

We will compare these results with our reference experiments and clarify the differences in the text.

*L325: initial estimates for the model parameters?*

We will revise it as suggested.

*L328: correlation between both parameters*

We meant the "establish spatial correlation for each parameter". We will revise the text.

*L332: what do you mean by "initial ensembles"*

We meant "initial model ensembles". We will revise the text accordingly for clarity.

*L354: need to fill in values for XX*

We will add values.

*L358: transient changes in model state but not in parameters*

We will revise it as suggested.

*L366: can you speak to the limitations on this? would having 100m resolution data be even better or proportionally so?*

In principle, higher-resolution data (e.g., 100 m) could further improve data assimilation performance by providing finer spatial detail on surface features and more precise constraints on model parameters. However, the benefit of finer resolution may decrease beyond a certain threshold due to increased observational noise, modeling uncertainties, and the inherent spatial correlation scale of the parameters being estimated. We will clarify this in the revised manuscript and discuss the potential limitations of very high resolution observations in the context of our OSSE framework.

*L370: this is a very important point that is worth highlighting in the abstract*

We will include this finding in the abstract.

---

## Author Response (AR1)

- Estimation of the state and parameters in ice sheet model
- using an ensemble Kalman filter and Observing System
- 3 Simulation Experiments
- Authors' response (RC1) -
- 5 Youngmin CHOI et al.
  - August 8, 2025
- 7 General comments

6

- 8 The manuscript by Choi et al. presents a data assimilation framework to improve the projection
- capabilities of ice sheet models. Specifically, the performance of an Ensemble Adjustment Kalman
- Filter in constraining the model state (ice thickness) and basal conditions (basal friction coefficient
- and bed topography) of a 2D plan-view ice model is assessed. Their results indicate that assimi-
- lating more observations generally increases the accuracy of model projections, with projections
- 13 for up to 200 years in close agreement with the reference simulation. The performance of the data
- assimilation method is sensitive to the observational error as well as the cross-track spacing and
- 15 grid resolution of surface elevation data.
- 16 I believe the science behind this study is sound and aligns with the focus of The Cryosphere (TC).
- 17 However, the presentation of the methodology lacks clarity, at times adding avoidable confusion
- 18 (e.g., the introduction of both acronyms EnKF and EAKF). This overall issue is addressed in more
- 19 detail in the specific comments below, but I strongly suggest the addition of a flowchart outlining
- 20 the methodology (ice sheet model and data assimilation) and experimental design (twin experiment
- 21 and OSSEs).
- We thank the reviewer for reviewing the manuscript and constructive comments. We address spe-
- 23 cific comments below as clearly as possible. We appreciate the suggestion to include a flowchart
- outlining the methodology and added it into the revised manuscript (Figure 1).
- 25 Furthermore, parts of the experimental design are currently placed within the results section and
- 26 key aspects of the results are not addressed (e.g., why is the RMSE C for Grid\_20 km $_{\sigma}h_{_{-}}20_{_{-}}\sigma_{v_{-}}10$

- smaller than for the same grid resolution with smaller uncertainties as well as all 10 km grid resolution experiments?). Considering the performance of the data assimilation method is sensitive to the uncertainty in surface elevation observations, I believe also determining the effect of various uncertainties in the velocity data would add further value to the manuscript (perhaps as supplementary material). Finally, potential reasons/explanations for model results are often missing, e.g., why is the range of optimal localization radius (4 8 km) a lot smaller than suggested by previous studies (4 120 km)? I recommend the authors also take my specific comments listed below into account.
- We agree with the reviewer that the some of the results lack sufficient explanation. In the revised manuscript, we added more explanation for our results by addressing specific comments listed below. Also we added additional experiments varying uncertainties in the velocity data in the appendix.
- 39 Specific comments
- L25: This sentence is very similar to the second sentence in the introduction. Instead, consider opening with a sentence about the different DA methods (variational vs. methods leveraging timevarying observations). Then proceed to discuss advantages/disadvantages of each.
- We revised this sentence. Now it starts with "Data assimilation (DA) methods for ice sheet modeling generally fall into two categories: snapshot and transient inversions, which use single-time observations and time series of observation, respectively." and continues to explain two methods.
- 46 L28: Double brackets.
- 47 Fixed.
- 48 L33: Consider starting a new paragraph before Alternatively.
- We revised it to start a new paragraph here.
- L37 assimilation period: Readers unfamiliar with DA might not know what exactly you refer to here. It becomes a lot clearer later on, but it would be nice to have a brief definition here (similar for other DA-specific terms, e.g., data denial experiments in L61).
- We added brief definitions of several DA terms for clarity, including the assimilation period and data denial experiments.
- 55 L40: Move further up to the rest of the discussion on variational methods.

- We revised this sentence to include the both static and transient methods.
- L42: Consider introducing the term ensemble DA in general before describing the specific EnKF (e.g., ensemble DA vs. variational methods).
- We added an introductory sentence for the ensemble DA in general.
- 60 L54: Why are ensemble DA methods less commonly used in ice sheet modelling?
- We added the following new paragraph to the revised manuscript in response to this point:
- "The ice sheet modeling community has traditionally relied on snapshot inversion methods based 62 on adjoint-based techniques for parameter estimation, using time-invariant mosaics or composite data (e.g., multi-year averaged surface velocity fields; Morlighem et al., 2010). Compared to these methods, ensemble DA approaches have been less commonly used in ice sheet modeling, primarily due to historical limitations in observational data, computational cost, and the challenges of representing uncertainty in ice sheet models. Ensemble approaches rely on time-varying obser-67 vations with well-characterized uncertainties, but surface observations for ice sheets have often lacked reliable uncertainty estimates, making them less suitable for ensemble DA. Additionally, ensemble methods typically require multiple forward model runs, making them more computationally demanding than snapshot inversion approaches. Another limitation is that poorly understood 71 or unquantified errors in the ice flow model itself may limit the reliable estimation of covariances 72 using ensemble statics."
- 74 L63 (OSSEs)(OSSEs, ...).
- 75 Fixed.
- 76 L65 appropriate observation error distribution: How do you determine if the distribution is appropriate or not?
- We revised this to "a prescribed observation error distribution representative of real measurement uncertainties".
- L71: Although it is addressed in more detail in the next sentence, I believe adding (ice thickness) just after model state would add clarity.
- 82 Added.
- L75 estimated state and parameters: For consistency, I recommend using estimated model state and parameters throughout the manuscript.

- Now, we use consistent terminology (model state) throughout the manuscript.
- L75 true reference values: At this point in the manuscript, it is not clear what the true reference values are and how you obtain them.
- 88 We revised it to "true values".
- L83-84: Remove this sentence and add the reference to the description of the specific sections in the text above.
- We removed this sentence and added reference to the description of the specific sections in the text.
- 22 L86-88: Repetition of the text just before the Methods section.
- 93 We revised this paragraph as follows:
- "This section describes the ice sheet model configuration (Section 2.1), the ensemble DA framework (Section 2.2), and the experimental designs used in this study (Section 2.3 and 2.4). We first outline the twin experiment setup, which tests the ability of the DA framework to recover the model state and parameters under idealized conditions. We then describe the OSSEs, which explore the effects of different observational strategies on model initialization. Our methods are summarized in Figure 1."
- L101: I am not familiar with this specific method, but a standard deviation of 500 m seems quite large considering the bed varies only between zb, deep = -720 m and  $\sim$ 500 m (really difficult to see in Fig. 1a)
- The midpoint displacement method generates a 2D surface by iteratively subdividing a grid, assign-103 ing random heights to corner points, and interpolating midpoints with added random displacement. 104 The magnitude of the displacement is scaled by a standard deviation that decreases with each iter-105 ation as  $2^{0.5H}$ , where H is the roughness factor, set to 0.7 in this study. While the initial standard 106 deviation of 500 m may seem large relative to the vertical range of the bed topography, it is used 107 as a starting point in the midpoint displacement algorithm and is progressively reduced at each 108 iteration based on the roughness factor. This results in a realistic, spatially correlated roughness 109 pattern with limited high-amplitude variations. Additionally, the current value of  $z_{b,deep} = 720m$ 110 represents the base shape of the bed before roughness is added and the final bed elevation reaches 111 depths of approximately -1,500 m. We added these details to the text and revised the Fig. 1a as 112 suggested below. 113

Fig. 1: I believe using two separate 2D plots instead of the 3D plot would make the identification of certain details and interpretation of the plot a lot easier. As you are already showing the bed topography in Fig. 2, I recommend combining Fig. 1 and 2 into a single plot with panels a: ice surface elevation, b: bed topography, and c: ice velocity. Note that the rainbow colour scheme is not in line with the journal guidelines. You can check all of your plots with the colour blindness simulator (https://www.color-blindness.com/coblis-color-blindness-simulator/). The fonts in panel b are too small and I recommend using a different colour for your contour lines.

We revised this figure as suggested.

Fig. 2: Why are you using such an asymmetrical (about y=40 km) bed topography compared to the commonly used symmetrical approach in idealized studies? The y label in Fig. 2 indicates the domain ranges from 20 to 100 km, whereas in Fig. 1 it is 0 to 80 km. You also might want to remove the white margins at the top and bottom. How can the bed elevation be -1500 m when Eq. 1 limits the bed to  $z_{b,deep} = -720$  m?

Applying the midpoint displacement method results in an asymmetrical bed topography, which may better reflect realistic subglacial features, although we use an idealized twin experiment in this study. We included this explanation and revised the figure as suggested.

- L103: Consider adding an additional panel (d) showing the triangular mesh to the new Fig. 1.
  Are you using adaptive mesh refinement, e.g., following the grounding line?
- We used an adaptive mesh based on ice velocity and included a new figure (Fig. 1(d)) in the revised manuscript.
- 134 L104: I suggest adding another panel (e) for the basal friction coefficient to Fig. 1 or at least refer 135 to Fig. 6a here.
- We added a reference to Fig. 6a to avoid repetition.
- Eq. 5: In case you are working in LaTeX, I recommend using left( and right) to get brackets of the correct size.
- 139 Fixed the bracket.
- 140 *Eq.* 6: Same as for Eq. 5
- 141 Fixed.

- L114: Is C in Eq. 7 different from the one described in Eq. 4? If not, then remove C is a friction coefficient.
- C is the friction coefficient, and  $C_x$  and  $C_y$  are the x and y components of C, respectively. We added this to the text.
- 146 L117: Remove equal.
- 147 Removed.
- L124: Do you consider a melt rate of 200 m/yr realistic given that maximum present-day melt rates are around 100 m/yr?
- We set the melt rate to 200 m/yr at a depth of 800 m, which results in an actual melt rate of approximately 170 m/yr beneath the ice shelf. We agree that this melt rate exceeds the maximum observed present-day basal melt rates. However, in this study, we chose this value to create a strong dynamic response in the model over a 200-year forecast period, ensuring that the effects of data assimilation could be clearly evaluated. The elevated melt rate is not meant to represent a realistic present-day climate, but rather to serve as a diagnostic tool in the context of a twin experiment. We clarified this in the revised manuscript.
- 157 L125-127: This belongs into results.
- This describes the process of creating the reference run for the twin experiment rather than presenting model results. We revised the text to clarify it.
- 160 L129: Sec. 2.3 and 2.4 are referenced before 2.2.
- We deleted this sentence.
- L134: modified version of the Ensemble Kalman Filter? As I mentioned above, using EnKF and EAKF is confusing, especially since EAKF is introduced but only used within this paragraph.
- We revised this paragraph to clarify the use of data assimilation terminology throughout the manuscript.
- 165 L136-143: This is where I think a flowchart would really help the reader to follow the details of 166 your method. Ideally, the flowchart should outline the details of the EAKF and how it relates to 167 your specific study. For example: How do ensemble members differ? What exactly is your model 168 forecasting? How is the observation window specified? Which ice sheet variables are considered 169 in the state vector? What are the state variables?

- We added a flowchart (Figure 1) and a reference to that in this paragraph.
- 171 L142: How would adding extra variables, like surface velocity, to the state vector affect your results?
- Surface velocity is a diagnostic variable; it is not part of the state vector. We added this in the revised manuscript as follows:
- "The state vector is augmented to include both prognostic variables and model parameters to be estimated. Under the stress balance of SSA, the velocity is a diagnostic variable, and due to the flotation condition, ice thickness is the only prognostic variable (Gillet-Chaulet, 2020). In this study, the state vector includes variables ice thickness (state variable), and basal friction coefficient and bed topography (model parameters), allowing joint estimation of the model state and parameter fields through the DA process (Fig. 1)."
- 181 L143-144: EnKFs or EAKFs? Does this challenge arise in your study? What is the ensemble size?
  182 What are the independently observed degrees of freedom in your case?
- Here, we explain the general case for the ensemble Kalman filter. We revised this paragraph to improve clarity and better distinguish the general description from our specific implementation.
- 185 *L146: stability of the EnKF?*
- 186 We changed it.
- 187 L150: Add more detail about what exactly you mean by sampling errors.
- We added "Sampling errors occurs because the ensemble based covariance is only an approximation of the true covariance, and small ensembles may not adequately capture variability across the full state space" to clarify.
- 191 L151: What localization and inflation parameters are you examining?
- We added a range of both localization radii (2 to 20 km) and inflation factors (1.00 to 1.20).
- Sec. 2.3: You either need to embed this information into the previous section or clearly outline at the beginning of the methods section that you are first describing the EnKF in general and then how this general structure relates to your specific setup (with references to sections). Again, a flowchart linking the general structure to your experiments would be helpful.

- We outlined the description of the methods at the beginning of the Methods section with a flowchart.
- 198 L154: EnKF or EAKF? If EnKF, then why bother introducing EAKF?
- EnKF is the general term, and EAKF is the specific approach we use in this study. We clarified this in the revised manuscript.
- 201 L164: Why did you decide to use lower standard deviations? What is the plausible range?
- We selected lower standard deviation values (5 m for surface elevation and 10 m/yr for velocity) 202 to provide a simple and conservative baseline for the twin experiment. While these values are 203 lower than those used in Gillet-Chaulet (2020), they are still within the plausible observational 204 uncertainty ranges reported in recent literature. For example, Dai and Howat (2017) report vertical 205 elevation uncertainties below 5 m in well-constrained regions, and Mouginot et al. (2017) report 206 horizontal velocity uncertainties ranging from 5–20 m/yr depending on the region. We chose values 207 at the lower end of these ranges to isolate the performance of the DA framework under favorable 208 conditions, and we explore sensitivity to larger uncertainties in the OSSEs presented in Section 3.2. 209 We clarified this in the revised manuscript.
- **.**
- 211 L178-179: Are you assuming that the friction coefficient and bed topography are uncorrelated?
- While we do not prescribe a prior correlation between them, the EAKF uses the ensemble-based cross-covariance to update both fields during the assimilation process. We included this explanation in the revised manuscript.
- 215 L184: What radii did you explore?
- The radii explored in the experiment ranged from 2 km to 20 km. We added this information to the text.
- 218 L186: Ensemble size of 30 to 100, but what steps exactly?
- We tested ensemble sizes of 30, 50, and 100, and specified this information in the revised text.
- Fig. 3: The font size is too small. This is generally the case for a lot of plots and I will refrain from mentioning it again afterwards. Otherwise, I think this is a great figure supporting the description of your OSSEs.

- We increased the font size for figures.
- L221 reference model mesh: Do you mean the mesh used in the reference simulation?
- Yes, we clarified this in the text.
- 226 L230-231: This information needs to come earlier.
- 227 We moved this information to the Method section.
- L234: Can you provide any insight as to why these values lead to the minimum RMSEs?
- The localization radius is determined through a set of sensitivity experiments and is based on the expected spatial correlation length scale of the parameters, which may depend on the size of flow features or stress balance regimes. We added some discussion on this point in the revised Discussion section.
- 233 L238: Why is that expected?
- Larger ensembles generally provide better approximations of the true error covariance, reducing the need for artificially inflating the covariance to compensate for sampling errors. We clarified this explanation in the revised manuscript.
- 237 L240: You are using inflation parameters in the text but inflation factor in Fig.4.
- 238 We revised it to "inflation factor" for consistency.
- 239 L243-245: For ensemble size 100, the RMSE for friction coefficient does NOT continue to decrease 240 steadily (increase at t=7a). The other two ensemble sizes also show a small increase just after 5 241 years. Similar peaks are also visible for the bed topography and ice thickness. What is causing this 242 increase in RMSE?
- We examined the small increase in RMSE in early assimilation years and found that it is likely due to a temporary mismatch between the model forecast and the observations during this period, potentially caused by transient model dynamics or nonlinearities in the response to assimilated observations. As the assimilation continues, the filter gradually corrects these discrepancies, which leads to a subsequent reduction in RMSE. These fluctuations are not uncommon in ensemble data assimilation systems, especially in complex, nonlinear models where localized error growth can temporarily degrade performance. We added this behavior in the revised manuscript (Discussion).

L245: A larger ensemble size (100 vs. 50) actually leads to a larger RMSE after t=15 a for all panels in Fig. 5. Why do you think this is the case? And what does it mean for the design of future experiments?

While larger ensemble sizes generally improve the accuracy of error covariance estimates, they can 253 also increase the sensitivity of the filter to model errors or sampling noise if not properly tuned. 254 In our experiments, it is possible that the inflation and localization parameters used for the 100-255 member ensemble were not optimal for later assimilation periods, leading to slightly degraded 256 performance after year 15. This suggests that filter performance does not necessarily scale linearly 257 with ensemble size. It also emphasizes the importance of adaptive inflation/localization techniques 258 or diagnostics for dynamically adjusting filter settings. We added this in the Discussion section of 259 the revised manuscript as a new paragraph: 260

"Larger ensemble sizes could improve data assimilation performance but may also introduce challenges that must be carefully managed, particularly in long assimilation periods or highly nonlinear systems, as in this study. In our experiments, it is possible that the inflation and localization parameters used for the 100-member ensemble were not optimal for later assimilation periods, leading to slightly degraded performance after year 15. This suggests that filter performance does not necessarily scale linearly with ensemble size and highlights the importance of adaptive inflation/localization techniques or diagnostics for dynamically adjusting filter settings."

L248: Why did you choose a localization radius of 4 km when bed topography and ice thickness showed a minimum RMSE for 6 km with a significant increase for smaller radii?

The minimum RMSE for friction occurs at a localization radius of 4 km and this radius produces decent results for bed topography and ice thickness as well. We chose 4 km for illustrative purposes and added this in the text.

273 L250: Somewhere you should state explicitly that a localization radius of 4 km and an inflation 274 parameter of 1.12 are your optimal DA configuration.

The optimal inflation factor and localization radius depend on the parameter being estimated. We chose 4 km and 1.12 for illustrative purposes and clarified this in the text.

L254-256: You describe the results for the friction coefficient and bed topography, but what about the ice thickness?

The pattern in the ice thickness results is very similar to that of bed topography. In our model setup, surface elevation is defined as the sum of ice thickness and bed topography (surface = thickness + bed). Therefore, as surface observations are assimilated, improvements in bed estimates are reflected in the estimated thickness field. We clarified this relationship in the revised manuscript.

L257-262: This is a description of your experimental design and should be in the methods section. 283

We moved this to the Methods section.

284

285

L264: Just to make sure I understood it correctly. The deterministic forecast uses the mean 286 (across all ensemble members) basal friction coefficient and mean bed topography but is a single 287 simulation (compared to running a simulation for all ensemble members and then calculating the 288 ensemble mean). 289

Yes, we clarified this in the text. 290

L268-269: I think it is quite interesting that the deterministic forecast, which is based on the 291 ensemble mean basal conditions, follows the reference simulation relatively closely while most of 292 the individual ensemble member simulations show a much smaller ice volume change. I suspect 293 this is due to non-linearities in the system, but it might be worth having a closer look at this.

We agree with the reviewer's point. While each ensemble member represents a physically plausible realization of the basal parameters, small deviations from the true field can lead to large differences 296 in modeled ice volume due to non-linear feedbacks. However, the deterministic forecast, initialized 297 with the ensemble mean of the basal fields, appears to capture the overall structure of the true 298 conditions more effectively, reducing local extremes and yielding results that are more closer to the 299 reference simulation. We added this discussion to the revised manuscript (Discussion). 300

L271-273: So assimilating more observations leads initially to a better agreement but increases 301 the difference in ice volume change at the end of the forecast period. This should be addressed in 302 more detail in the discussion. 303

Assimilating more observations leads to better agreement throughout the forecast period, including 304 at its end. We included specific values for ice volume loss to support this comparison in the revised 305 manuscript. 306

L273-274: Discussing the increase in the rate of mass loss after 100 a and its implication for 307 sea level rise projections over the next century (compared to 200 yr projections) could be a nice 308 additional takeaway. 309

We added this to the Discussion section as follows: 310

"Notably, the 200-year reference simulation includes a phase of accelerated volume loss after 311 130 years, which may represent a plausible sea level rise scenario for the coming century. Our

- results suggest that assimilating observations even before such nonlinear transitions can still reproduce accurate long-term projections—provided that the model state and parameters are well constrained."
- Fig. 4: Why are some experiments diverging? Why does the diverging area shift to smaller inflation factors as the ensemble size increases? What is the reason for the sharp increase in RMSE for localization radii smaller than the minimum RMSE? Use friction coefficient, bed elevation, and ice thickness as labels next to the colour bar.
- When the localization radius is too small, it overly restricts the influence of observations on the state update. This can lead to underestimation of error covariances across space and result in filter divergence. In our experiments, this is evident when the localization radius falls below the specific threshold of each variable (e.g., 4 km for friction and 6 km for bed topography). We included this explanation in the revised manuscript (Discussion). We also revised the figure as suggested.
- Fig. 5: I suggest using RMSE\_C, RMSE\_B, and RMSE\_H as y-labels or adding friction coefficient, bed elevation, and ice thickness as titles. The description of panel c is missing. The colour coding is somewhat confusing because you are using the same colours as in Fig. 4 but they do not represent the same thing (ensemble size vs. friction coefficient/bed elevation/ice thickness).
- We revised the figure as suggested.
- Fig. 6: Why did you choose a more or less symmetrical (about y=40 km) friction coefficient but a very asymmetrical bed topography? The units are missing in the colour bar label. You might want to increase the spacing between panels to make it clearer which text corresponds to which panel.
- 333 We revised the figure as suggested.
- Fig. 7: What causes the sharp grounding line extent towards higher x values at y=10 km in the no assimilation panel? Units are missing. Increase spacing between panels.
- We revised the figure as suggested.
- Fig. 8: Why did you not show the difference in Fig. 6 and 7? What causes the checkerboard pattern?
- We chose to show the difference in ice thickness in Fig. 8 because changes in thickness are difficult to detect visually from the similar figure as Fig. 6 and 7. The artifacts observed in the ice thickness are the result of the conditional random fields generated using the Kriging method, which can

- produce "bull's eye" patterns commonly observed between observation points. We clarified this in the revised manuscript.
- Fig. 9: I suggest adding a legend for the different lines. Change reference run to reference simulation. Change forecast simulation to deterministic forecast simulation (all of them are forecast simulations).
- We revised the figure as suggested.
- Sec. 3.2: Do not use abbreviations as section titles.
- 349 We changed the title.
- 250 L285: Change 10 to 15 km to 15 km. Or did you also test, e.g., 13 km across-track spacing?
- We revised this as suggested.
- L286: I am not sure what you mean by the performance declines due to suboptimal choices for inflation and localization parameters. You are examining the effects of these parameters here, so shouldn't you be able to determine the optimal choices? Do you mean the optimal choices are
- outside your tested parameter ranges? If so, you need to show results supporting this claim.
- We removed the phrase "suboptimal choices for parameters".
- L287: Start a new paragraph before "For the gridded".
- We revised it as suggested.
- L288: Add the reference to Fig. 11 to the end of this sentence (currently at the end of the paragraph).
- We revised it as suggested.
- 362 L289: range of localization.
- 363 Done.
- L292: we conducted: Use present tense throughout the manuscript (e.g., same issue in L403 presented).

- We revised the manuscript to consistently use the present tense.
- Fig. 10: In panels (f) and (i), why is the RMSE of localization radius 6 km and inflation factor 367 1.04 much larger than the surrounding values? 368
- The RMSE values are calculated over the entire domain, and localized errors can increase the 369
- overall RMSE. We included additional discussion on the localization and inflation factors in the 370
- revised manuscript. 371
- L297-299: Add reference to Table 2. 372
- We added the reference to Table 2.
- L301: RMSE values continue to decrease until the end: Add reference to Fig. 12. Panels c, f, and 374
- i show an increase in RMSE at the end.
- We added the reference to Fig. 12 and an increase in RMSE. This paragraph is about when the 376
- elevation error is 5 m/yr which does not an increase in RMSE at the end.
- L301-303: Add reference to Table 3. 378
- We added the reference to Table 3. 379
- L304: marginal improvements ... after 10-15 years: Add reference to Fig. 13. Again, RMSE 380 actually increases in panels f and i. What do you think causes this increase? 381
- We added the reference to Fig. 13 and described the increase in RMSE for the 20 km grid data. We 382 also added some discussions on this point in the revised manuscript. 383
- L304-305: This is discussion. 384
- We remove this sentence here since there is already a similar discussion point in the Discussion 385 section. 386
- L307: Add reference to Table 2. For RMSE\_C, the smallest uncertainty leads to the second-largest 387 RMSE of all tested uncertainties. So DA performance does not necessarily decrease as uncertainty
- increases!

- We agree with the reviewer that this paragraph is not well presented. We revised it to clearly reference Table 2 and Fig. 12 and distinguish between the RMSE values for friction coefficient (RMSE\_C) and bed topography (RMSE\_B) and ice thickness (RMSE\_H).
- L308-309: Add reference to corresponding panel in Fig. 12. For the 10 km across-track data, the maximum difference in RMSE\_C is 15.43. For the 5 km data, it is only 8.65. Although the maximum difference is much smaller in the 5 km case, you argue that it shows a decrease in performance as uncertainty increases while the larger 10 km difference indicates a consistent performance across different uncertainty levels. I believe your statement is primarily based on the RMSE\_B and RMSE\_H results, but these details need to be spelled out!
- We added reference to corresponding panel in Fig. 12. In addition, we clarified the distinguish between the RMSE values for friction coefficient (RMSE\_C) and bed topography (RMSE\_B) and ice thickness (RMSE\_H) in the revised manuscript.
- L309-311: Adding to my previous comment, if you compare  $Track\_15km\_\sigma_h\_10\_\sigma_v\_10$  to  $Track\_15km\_\sigma_h\_15\_\sigma_v\_10$ , and  $Track\_15km\_\sigma_h\_20\_\sigma_v\_10$ , the DA performance increases as uncertainty increases for RMSE\\_C, RMSE\\_B, and RMSE\\_H. This needs to be stated clearly and discussed in detail!
- 405 We revised this paragraph to clarify.
- L311: Add a reference to the corresponding panel in Fig. 12. They actually show an increase in RMSE after 15 to 20 years.
- We revised this paragraph to add this point.
- 409 L311: Add new paragraph before "With the 1 km gridded".
- 410 As suggested, we separated the discussion of the 1 km gridded results into a new paragraph.
- L312: Add a reference to Table 3. What about the friction coefficient? For the friction coefficient and Track\_15km, the highest uncertainty level has the smallest RMSE and, therefore, the best performance. So DA performance does not necessarily decrease as uncertainty increases!
- Although the highest uncertainty level has the smallest RMSE, the lowest uncertainty level still has smaller RMSE than the 10 m/yr and 15 m/yr uncertainty results. In addition to the RMSE pattern during the assimilation window (Fig. 13a), we argue that the friction coefficient does not show the clear pattern with varying uncertainty in surface elevation. We added this point in the revised manuscript.

- L313: DA performance does not vary significantly across different uncertainty levels: I don't 419 think I agree with this statement. Again, just looking at RMSE\_C, the maximum difference across 420 uncertainty levels at 1 km resolution is 2.84, while it is 10.28 at 10 km and 30.8 at 20 km. So if 421 anything, the performance varies more significantly with coarser grid resolution. Additionally, the 422 RMSE\_C for the coarsest grid resolution (20 km) and highest uncertainty level is smaller than all 423 other RMSE\_C values at 20 km AND 10 km resolution! As the RMSE\_C with the highest uncertainty 424 and the coarsest across-track resolution is also smaller than all other values at this resolution, it 425 seems unlikely that this is just a coincidence. So this really needs to be addressed. 426
- We apologize for the errors in the RMSE\_C values for the 20 km grid data (caused by copy-and-paste mistakes in the latex document). We corrected these values in Table 3 and revised this paragraph accordingly.
- Fig. 12: Use RMSE\_C, RMSE\_B, and RMSE\_H in y-labels. Panel labels in the second and third rows are the same. Actually, even the subplots themselves look the same. The panel labels seem to be just a copy/paste issue in your Python code, not sure about the actual data.
- We fixed the panel labels. The pattens for RMSE\_B and RMSE\_H over assimilation time is very similar to each other since surface elevation is defined as the sum of ice thickness and bed topography (surface = thickness + bed). The figures in the second and third lows look very similar but not the same figures (note that values in y axis).
- Fig. 13: Same issues as for Fig. 12. What causes the increase in panel f after 20 years? Why is there such a rapid increase in RMSE in, e.g., panel d between 5 and 10 years? Why does this rapid increase get muted for coarser resolutions? A similar pattern occurs in Fig. 12.
- As mentioned above for other figures (e.g., Fig. 5), it is likely due to a temporary mismatch between the model forecast and the observations during this period, potentially caused by transient model dynamics or nonlinearities in the response to assimilated observations. Localized error growth can temporarily degrade performance. We included discussion on this point in the revised manuscript.
- L316: fast flowing regions: You haven't mentioned fast-flowing regions before. Are you referring to areas around the grounding line, where the signal-to-noise ratio of velocity is relatively high? I'd argue that large differences also occur for y=70-80 km and x=450-640 km, which seems to be a relatively slow-flowing region.
- We specified fast flowing regions as where velocity is larger than 100 m/yr, which corresponds to the around the grounding line in the revised manuscript.
- L319: fast flowing regions: Again, what about the region between y=70-80 km and x=450-640 km?

- We specified fast flowing regions above.
- 453 *L322: "more" accurate.*
- 454 Done.
- 455 L333: assumptions on the initial ensemble: Be more specific.
- We revised this to "how the initial ensemble is generated".
- 457 L337: relatively small ensemble size: Be precise.
- We specified the ensemble size.
- 459 L337-338: previous studies: References are missing.
- We added the references.
- L342: a larger ensemble size could provide advantageous: Or not. Fig. 5e shows a larger RMSE for 100 members than 50 members by t=30 years.
- We agree that increasing ensemble size does not always guarantee improved DA performance.
- We revised the discussion to acknowledge that larger ensemble sizes can improve performance in
- general but may also introduce challenges that must be carefully managed, particularly in long
- assimilation periods or highly nonlinear systems.
- 467 L343: What exactly should these studies investigate to identify the optimal approach?
- We removed this sentence, as paragraphs discussing future studies are now included only at the end of the discussion section, as suggested by the reviewer.
- 470 L345: similar to values from earlier studies: References are missing. What are these values?
- We added the references with values from those studies.
- 472 L347: 4-120 km is a lot wider range than your 4-8 km. What causes these differences?

- The differences in the optimal localization radius likely comes from the differences in model con-
- figuration, dimensionality, and spatial resolution. Our study uses a 2D unstructured mesh with
- relatively fine spatial resolution, whereas previous studies using flowline models (1D) with coarser
- grids may require broader localization to account for longer correlation length scales. We added
- this point in the revised manuscript.
- 478 L351: ... assimilating more observations, i.e. more assimilation years, to estimate ...
- We revised it as suggested.
- 480 L352: improves accuracy of model projections: In general, yes, but the difference in ice volume
- change at t=200 years is larger in Fig. 9 panel b than panel a (between red and blue line).
- We added "reduced uncertainty" to specify the accuracy. As the reviewer mentioned, in general
- and during the projection period, the determinist ice volume loss forecast in Fig. 9b shows better
- agreement with the reference simulation than in Fig. 9a.
- 485 *L354: XX : Add numbers.*
- 486 Added.
- 487 L358: in this study that: Replace with here.
- We revised this as suggested.
- 489 L358: observations maintains: Replace with observations while maintaining
- We revised this sentence.
- 491 L361: I suggest restating what OSSE means for readers quickly skimming through the manuscript.

492

- We added a brief definitions of OSSEs.
- 494 L361: Remove "in this study".
- 495 Removed.
- 496 L361: the capabilities of OSSEs: Replace with their capabilities.

- 497 Done.
- L365: (Table 2 and 3): If you include references here then you should also include them in previous paragraphs of the discussion.
- 500 We removed these references.
- L365-366: As indicated previously, this is not the case for RMSE\_C in Table 3.
- Again, we apologize errors in RMSE\_C in Table 3. We revised this paragraph to clarify our points.
- 503 L369-371: As mentioned above, additional data points can actually have a negative effect.
- We revised this sentence.
- L373-374: I disagree. The friction coefficient has the largest differences in RMSE across uncertainty levels (for all resolutions in Table 2 and similar for Table 3), so it is actually the most sensitive.
- We intended to compare DA performance in estimating bed topography (and ice thickness) versus the friction coefficient. We revised it to "friction coefficient retrieval shows no clear pattern in response to the prescribed surface elevation uncertainty".
- 511 L377: will or should?
- 512 We revised it to "should".
- 513 L379-380: Consolidate all future study suggestions into one paragraph.
- We removed some mentions of future work, and now they are included in the final paragraphs of the discussion section.
- L398-399: Your future studies should address sentences are spread out across the entire discussion. I recommend bundling all of them into one single paragraph at the end of the discussion.

  518
- We removed some mentions of future work, and now they are included in the final paragraphs of the discussion section.

- 521 L398-399: Again, this is not always the case.
- 522 Except for the temporary decreases in DA performance observed during the assimilation period,
- this statement generally holds true. We clarified this point in the revised manuscript.
- 524 L400: What does great accuracy mean? Be precise.
- We agree with the reviewer and replaced "great accuracy" with a quantitative assessment.
- 526 L404: Different levels of observational uncertainty: Do you mean smaller levels of uncertainty?
- It is not necessarily smaller uncertainty levels; it also depends on the resolution or the track spacing of the data. We clarified this in the revised manuscript.
- 529 L410: Will you also upload the data files?
- Yes, we uploaded the data files to run the scripts.

- Estimation of the state and parameters in ice sheet model
- using an ensemble Kalman filter and Observing System
- 3 Simulation Experiments
- Authors' response (RC2) –
- Youngmin CHOI et al.
- August 6, 2025

6

- This is a review of "Estimation of the state and parameters in ice sheet model using an ensemble Kalman filter and Observing System Simulation Experiments" by Choi et al., submitted for publication to The Cryosphere. This manuscript describes the use of an ensemble-based data assimilation system, the Ensemble Kalman Filter (EnKF), to assimilate data into a 2D large-scale ice sheet models, for the purpose of better estimating parameter values and state variables during the historical period. It follows on other studies that have explored similar methods for 1D ice sheet models, and makes the crucial step of applying such methods to a model widely used for projections. This study also adds a novel "Observing System Simulation Experiment" in which different potential observing system configurations (resolution, track spacing, observational accuracy) are tested to determine their ability to improve accuracy in estimated parameters and state.
- Overall, I think this is a pretty straightforward study using well-known tools in a new way with ice sheet models, advancing the state of the art in our field. My main suggestions are to further explore certain DA and modeling choices that are unexamined in the current version of the manuscript. I have detailed these suggestions and more minor ones below.
- We thank the reviewer for reviewing the manuscript and constructive comments. We revised the manuscript to include additional justification for key data assimilation and modeling choices, clarify methodological decisions, and expand the discussion on the implications and limitations of our approach. We address each specific comment in detail below and aim to improve the clarity.
- 1. The manuscript briefly describes what the EnKF is, and then indicates that the EAKF version is chosen for this study. There are multiple different flavors of the EnKF available in DART, so it

is unclear why EAKF is chosen and whether the results would be any different if another filter was chosen. My suggestion is to describe in some more detail what is done in an EnKF and how the EAKF is different from the standard EnKF. Additionally, either some justification for why the EAKF was chosen and some level of justification for why that is the preferred approach when others are available.

Thank you for the suggestion; this was also raised by another reviewer. We added more details about the EnKF and EAKF, and the description about the distinctions between the two approaches in the Data assimilation section as follows:

"In contrast to the standard stochastic EnKF—which perturbs observation-space quantities randomly for each ensemble member to account for observational uncertainty—the EAKF avoids additional perturbations and instead analytically adjusts the ensemble members to match the posterior
mean and covariance determined by the original Kalman filter equations (Anderson, 2001). This
approach improves numerical stability and reduces sampling noise over stochastic EnKFs, especially for small ensemble sizes (Whitaker and Hamill, 2002). In this study, we choose the EAKF
due to its reduced sensitivity to ensemble size and improved robustness in geophysical systems, as
demonstrated in previous studies using DART (Zubrow et al., 2008; Anderson et al., 2009)."

2. One thing that is unclear from your study design is the relative importance of assimilation window (e.g. 5 vs 15 vs 30 years) as compared to number of assimilation cycles. You don't change the frequency of observations, which may be sensible given than annual observations are reasonable for current observing platforms. However, it is then hard to understand as a reader whether there is something fundamental about having 20-30 years of observations related to the time scales of ice sheet response to adjustments, or whether it is having 20-30 assimilation cycles to improve. If the observations were more frequent (e.g. an IceSAT2-like 90 days) would it takes less time for the EnKF to improve to the level that you show here?

This is a great point. In this study, we aimed to estimate two constant-in-time parameter fields and the model state on an annual basis. Given the timescales associated with the model state and parameters, as well as the capabilities of current observational platforms, we chose to use annual observations for simplicity.

We agree that the distinction between the length of the assimilation window and the number of assimilation cycles needs further investigation. To address this, we conducted an additional experiment using semiannual observations to explore the relative impact of the number of assimilation cycles versus the time period over which they are applied. We included the results and a discussion in the revised manuscript as follows:

"In this study, we focus on estimating two constant-in-time parameter fields and the model state using annual observations over assimilation windows of varying lengths (5, 15, and 30 years). This

choice is motivated by both the timescales associated with glacier dynamics and the current capabilities of observing platforms. However, the relative importance of the assimilation window length (i.e., total time span) versus the number of assimilation cycles (i.e., update frequency) remains an 64 open question. To explore this, we conduct an additional experiment using semiannual observa-65 tions under the same setup as the twin experiment (Fig. A1). The results suggest that semiannual observations lead to a faster reduction in RMSE for both the model state and parameters. However, 67 the improvement at the end of the 30-year assimilation window, compared to annual assimilation, remains limited. This limited benefit is likely due to the nature of the parameters and state vari-69 ables considered in this study—constant-in-time fields and annual-scale variability—which allow 70 sufficient information to accumulate over time for a fixed target. Once sufficient assimilation cycles 71 have passed, the parameters become well constrained, and more frequent updates offer little addi-72 tional improvement. These findings suggest that, for slowly varying or static variables, increasing 73 observation frequency can accelerate convergence toward the true state and parameter values, but 74 may not yield additional improvement beyond a certain number of assimilation cycles. In contrast, 75 if parameters or states change more rapidly or nonlinearly, a longer assimilation window or more 76 complex update schemes might be needed to achieve similar improvements. Future work should 77 explore the sensitivity of EnKF performance to both assimilation frequency and window length to identify optimal configurations for real glacier systems with time-varying parameters and limited 79 observation periods." 80

3. A big difference between your perfect model design and a real scenario where DA might be applied is that only two constant-in-time parameter fields are unknown. In reality, (e.g.) ice viscosity and climate forcing are also likely to be poorly known (though at least climate forcing is directly observable), and climate forcing (and basal friction) may vary in time. Two possibilities that would be helpful to run some experiments to assess are:

(a) if you are mistaken about the values of other parameters, but still only estimate basal friction
 and topography, will the estimate of basal friction compensate for these other errors (particularly
 for ice viscosity which trades off quite directly with basal friction in a depth-integrated model) there is some evidence of such compensation already happening in your estimates, see below

(b) could this DART-ISSM configuration be used to estimate multiple parameters at once (I don't see a reason why not, but the performance may not be the same as what is found for the single parameter estimation experiments explored currently).

I get that the design of these experiments are meant to mimic and compare directly to Gillet-Chaulet 2020, but it would be useful to also push beyond their design to get closer to a realistic case where DA might be used.

Thank you for this insightful comment. We agree that in real-world applications, other parameters such as ice viscosity and climate forcing are also poorly constrained and may vary in space and

time. In this study, we intentionally limited the number of unknowns to two constant-in-time parameter fields (basal friction and bed topography) to enable direct comparison with Gillet-Chaulet et al. (2020) and to isolate the behavior of the EnKF framework in a controlled setting.

Regarding (a), we recognize that compensation effects between parameters (e.g., between basal friction and ice viscosity) may arise if other sources of uncertainty are not properly accounted for.
We added a discussion of this limitation and clarified that our experimental design assumes perfect knowledge of other parameters such as ice viscosity, which is not realistic.

Regarding (b), while we agree that the DART-ISSM framework is capable of estimating multiple parameters simultaneously, we chose to start with a reduced set of unknowns to rigorously test the baseline performance of the ensemble Kalman filter. We believe this simplified setup is appropriate for the scope of the current study, which aims to provide a foundational assessment of ensemble DA performance in an ice sheet modeling context. Extending the framework to include more unknown parameters—such as ice viscosity or time-varying climate forcing—will likely introduce new complexities. These directions are indeed important and are part of our ongoing and future work, but they are beyond the scope of this initial system evaluation. We have clarified this in the revised manuscript as follows and note that this study provides a foundational step toward that goal by demonstrating the feasibility and utility of ensemble DA with a simplified setup.

"Our experimental design also assumed perfect knowledge of all model parameters except for basal friction and bed topography. This choice was made to facilitate learning about the DA system in a controlled setting and to keep the experimental setup more tractable, while also allowing for direct comparison with Gillet-Chaulet (2020). However, this approach limits the realism of experiments. In practice, parameters such as ice viscosity and climate forcing are also poorly constrained and may vary in both space and time. For example, uncertainties in viscosity may interact with basal friction during assimilation, potentially leading to parameter compensation effects. Future sensitivity studies should explore how mis-specified background parameters (e.g., biased viscosity fields) affect the estimation of other parameters and whether such compensation leads to biased or unstable forecasts. Although this study focused on estimating two constant-intime parameter fields (friction coefficient and bed topography), the DART–ISSM framework is well-suited for the joint estimation of multiple spatially or temporally varying parameters. Extending the current configuration to include additional unknowns – such as ice viscosity, accumulation rate, or time-varying boundary conditions – represents a valuable next step toward more realistic data assimilation in ice sheet modeling."

4. At more than one point it is suggested that using variational methods is more computationally intensive that ensemble-based DA. However, there is no real direct proof of this as you don't perform a direct comparison and to my knowledge this has not been done in the published literature. Given that ISSM has a variational DA option already implemented, it could be valuable to compare EnKF with ISSM to EnKF with ISSM in terms of core-hours for a simple standardized run. Short of that, it would be useful to have a sense for the DART overhead? If it is negligible then

I would expect ensemble-based DA to have n times the computational expense of a conventional ISSM run where n is the number of ensemble members. Additionally, giving a sense for how this ensemble-based approach has (or can be) parallelized would be useful. In theory, ensemble DA is highly amenable to parallelization, but this depends on how covariance matrices are constructed and how shared memory parallelism is handled. More details on all the computational aspects of this new method would be very useful to include.

Thank you for the great suggestion. We agree that a computational comparison between the two data assimilation methods would be valuable. However, due to the fundamental differences in the core computational processes of variational and ensemble based approaches, a direct comparison of their computational costs is challenging and is beyond the scope of this study.

Variational approaches using automatic differentiation (AD) tend to have higher memory demand,
while ensemble DA methods primarily increase computational cost through the need to run multiple forward simulations. Additionally, the two approaches are implemented using different tools
within ISSM – variational DA is built using the AD tool, while ensemble DA is integrated via
DART – which further complicates direct comparison. We have included this limitation in the revised manuscript as follows and mentioned the need for future work to systematically assess the
computational trade-offs between these two methods.

"Although ensemble-based data assimilation offers conceptual and practical advantages, its com-153 putational cost is often considered a limiting factor. In this study, we did not perform a direct 154 computational comparison between ensemble and variational (transient) DA approaches. Such 155 a comparison is challenging due to their fundamentally different implementations. For example, 156 variational DA in ISSM relies on automatic differentiation (AD), which can be memory-intensive, 157 whereas ensemble DA increases computational cost primarily by requiring multiple forward simu-158 lations. However, ensemble approaches can be parallelized, as each ensemble member's forward 159 run can be distributed across separate cores or nodes, and the DA process here is managed through 160 DART, which supports parallel computing. While formal benchmarking was beyond the scope of 161 this study, it would be valuable in future work to quantify computational trade-offs across DA 162 methods in ice sheet modeling." 163

164 L16: less numerical model re-development

165 Done.

166 L26: use a form of variational

167 Done.

168 L28: realy on observational at a single time to

- 169 Done.
- 170 L30: it often introduces nonphysical artifacts into the
- 171 Done.
- 172 L35: The use of computational techniques such as automatic differentiation in ice sheet models
- 173 Done.
- L69: to my knowledge this is the first ice sheet modeling paper to apply OSSE, so I think you can
- be more direct about this sentence
- 176 We revised this sentence as suggested.
- 177 L86: an ensemble...for ice sheet model initialization
- 178 We revised this paragraph.
- 179 L88: on model initialization
- 180 We revised this paragraph.
- 181 L92: simulation of ice sheets
- 182 Done.
- 183 L100: explain what the random midpoint displacement method is
- 184 We added the details on this method.
- 185 L138: model simulations
- 186 We revised this paragraph.
- L143: I am confused here because you don't include velocity in the state vector, but later you say it is part of what is assimilated?

- Velocity is an observation being assimilated, not the part of the state vector. We clarified this in the text.
- L184: does localization as implemented preserve covariance between different variables at the same location in space or does it simply localize along the diagonal of the covariance matrix? For example, there should be strong correlation between the ice thickness estimate and the bed topography estimate, and so you would be losing a significant amount of your ability to assimilate if covariances between these two variables at the same location in space where zeroed out by localization.
- Localization is applied to joint observation-state-space covariance in DART, which means that covariance across model variables at the observation location will not be damped by the localization. We added the following to the revised manuscript:
- "Localization is applied to reduce correlations between model states projected into observation space and the unobserved state variables, which does not explicitly damp covariances across colocated variables."
- L205: what would happen if you had no velocity observations? How much of the performance is due to velocity observations vs thickness?
- We acknowledge that this trade-off between observation types is important for real-world applications. As similar points are raised by another reviewer, we conducted additional experiments with varying uncertainties in the velocity data and included the new results in the revised manuscript. We also added a discussion on the relative importance of velocity versus elevation uncertainty.
- 209 L243: I think this should refer to Fig. 5
- Yes. We revised it to Fig. 5.
- 211 L259: mean to initialization the deterministics...full ensemble to initialization the ensemble
- We revised it as suggested and moved this paragraph to the Method section as suggested by another reviewer.
- Fig 4. In the caption you mention that highly localized experiments diverge. It would be helpful to speak to why these experiments diverge in the main text.
- We added this point to the Discussion section to clarify that when the localization radius is too small, it overly restricts the influence of observations on the state update. This can lead to underestimation of error covariances and result in filter divergence. In our experiments, this is evident when

- the localization radius falls below the specific threshold of each variable (e.g., 4 km for friction and 6 km for bed topography).
- Fig. 7: There are artifacts in the bed topography and ice thickness estimates that correspond to the basal friction estimate. Can you speak to this? Is it related to how the localization is performed?
- The artifacts in the bed topography and ice thickness are the result of the conditional random fields generated using the Kriging method, which can produce "bull's eye" patterns commonly observed between observation points. We added this to the Results section.
- L276: can you quantify this change in spread? by eye it doesn't seem to change much between 20 and 30 years of assimilation
- We added specific values for each spread in the revised manuscript.
- 229 Figure 9: can you add a legend and plot the ensemble mean as well?
- 230 We added a legend and included the ensemble mean in the plot.
- L290: It would help to discuss what this sentence means in practice. Is prediction accuracy degraded for this case? Or can you achieve similar results with different localization and inflation parameters?
- Although minimum RMSE values were identified for the coarser-resolution cases (10 km and 20 km), the overall prediction accuracy is still lower than that of higher-resolution cases (e.g., 1 km). We added this in the revised manuscript.
- 237 L325: initial estimates for the model parameters?
- 238 We revised this sentence including the suggestion.
- 239 L328: correlation between both parameters
- 240 We meant the "establish spatial correlation within each parameter". We revised the text.
- 241 L332: what do you mean by "initial ensembles"
- 242 We meant "initial ensemble of parameters". We revised the text for clarity.

- L354: need to fill in values for XX
- 244 Fixed.
- 245 L358: transient changes in model state but not in parameters
- 246 We revised it as suggested.
- L366: can you speak to the limitations on this? would having 100m resolution data be even better or proportionally so?
- In principle, higher-resolution data (e.g., 100 m) could further improve data assimilation performance by providing finer spatial detail on surface features and more precise constraints on model parameters. However, the benefit of finer resolution may decrease beyond a certain threshold due to increased observational noise, modeling uncertainties, and the inherent spatial correlation scale of the parameters being estimated. We added this in the revised manuscript.
- 254 L370: this is a very important point that is worth highlighting in the abstract
- 255 We included this finding in the abstract.

---

## Referee Report (RR1)

**Review of "Estimation of the state and parameters in ice sheet model using an ensemble Kalman filter and Observing System Simulation Experiments"**

**August 27, 2025**

The revised version of the manuscript greatly improved the presentation of the methodology and experimental design. It has a more logical flow, and, with the exception of some minor issues outlined in the specific comments below, the key aspects of the results are now adequately addressed and discussed. The additional experiments presented in the Appendix provide further value.

**Specific comments**

**[Now, we use consistent terminology (model state) throughout the manuscript.]**

There are still instances in which you do not use *model state* (e.g., L12).

**[L26-27]**

I suggest adding a sentence that better links the two paragraphs by describing how DA methods can reduce the uncertainty in key model parameters and model initialisation.

**[L52-54: As new observations are incorporated within the assimilation period, the ensemble mean presents an increasingly more accurate estimate of the model state.]**

This seems to contradict your statement in L14-15: additional observations do not improve and may even degrade long-term estimates of model parameters and state

**[We revised it to "true values".]**

I suggest adding "synthetic twin experiment" in L95.

**[We used an adaptive mesh based on ice velocity and included a new figure (Fig. 1(d)) in the revised manuscript.]**

Add a reference to Fig. 1d to the end of L127. Is the mesh adjusted as the grounding line retreats? Note that the rainbow colour scheme in panel d is not in line with the journal guidelines.

**$[z_{\text{b,deep}} = \text{Depth of the bedrock topography in Table 1}]$**

This is confusing, as your bedrock topography in Fig. 2 varies spatially. Consider *Maximum depth of the initial bedrock topography*.

**[L156: these surface and basal forcings]**

Since the lines directly above are discussing the melt rate, consider re-stating what these forcings are or refer to them as *perturbed surface and basal forcings*.

**[L169: we choose the EAKF due to its reduced sensitivity to ensemble size]**

This is somewhat confusing, as in L179, you state a common challenge with [...] EAKFs [...] arises when the size of the ensemble is significantly smaller than the independently observed degrees of freedom.

**[L282: EnKF]**

Should this be EAKF?

**[L297: continues to decrease steadily]**

It does not continue to decrease steadily.

**[L301-302: the benefits saturate as the ensemble size increases from 50 to 100]**

Be more precise here. For bed elevation and ice thickness, an ensemble size of 100 has a larger RMSE at t=30a than for a size of 50.

**[L302: illustrative purposes]**

What do you mean by that?

**[Fig. 7: What causes the sharp grounding line extent towards higher x values at y=10 km in the no assimilation panel?]**

You did not provide an answer to this question.

[We chose to show the difference in ice thickness in Fig. 8 because changes in thickness are difficult to detect visually from the similar figure as Fig. 6 and 7.]

Yes, but wouldn't it also be easier to detect the changes if you plotted the differences in Figs 6 and 7?

**[L362: although RMSE values continue to decrease until the end of the assimilation window]**

At 15 km across-track spacing, not all RMSE values continue to decrease.

[We added the reference to Fig. 13 and described the increase in RMSE for the  $20~\mathrm{km}$  grid data.]

Fig. is missing for some of these references.

**[L437: observations years]**

Change to observation years.

I hope the authors find my comments helpful.

Sincerely, Kevin Hank

---

## Author Response (AR2)

- Estimation of the state and parameters in ice sheet model
- using an ensemble Kalman filter and Observing System
- 3 Simulation Experiments
- Authors' response (RC1) -
- Youngmin CHOI et al.
- September 23, 2025
- 7 The revised version of the manuscript greatly improved the presentation of the methodology and
- 8 experimental design. It has a more logical flow, and, with the exception of some minor issues
- 9 outlined in the specific comments below, the key aspects of the results are now adequately addressed
- and discussed. The additional experiments presented in the Appendix provide further value.
- We thank the reviewer for their constructive comments, which have helped improve the manuscript.
- We address the minor issues below.
- [Now, we use consistent terminology (model state) throughout the manuscript.]
- 14 There are still instances in which you do not use model state (e.g., L12).
- 15 We have corrected all remaining inconsistencies and now use model state throughout the manuscript.
- L26-27: I suggest adding a sentence that better links the two paragraphs by describing how DA
- methods can reduce the uncertainty in key model parameters and model initialisation.
- 18 We have added the following sentence at the beginning of the second paragraph to improve the
- 19 flow:

6

- 20 "Data assimilation (DA) is a method of combining information from models with observations to
- 21 improve the accuracy of the model state variables and/or specific model parameters."

- L52-54: As new observations are incorporated within the assimilation period, the ensemble mean presents an increasingly more accurate estimate of the model state.
- This seems to contradict your statement in L14-15: additional observations do not improve and may even degrade long-term estimates of model parameters and state.
- This sentence (L52-L54) is meant to describe the general expected behavior of ensemble data
- 27 assimilation methods. However, our results show that DA performance can degrade under certain
- condition. We have revised the sentence to:
- 29 "As new observations are incorporated within the assimilation period, the ensemble mean is gen-
- 30 erally expected to provide an increasingly more accurate estimate of the model state under ideal
- 31 conditions."
- We revised it to "true values".
- 33 I suggest adding "synthetic twin experiment" in L95.
- 34 Done.
- We used an adaptive mesh based on ice velocity and included a new figure (Fig. 1(d)) in the revised manuscript.
- Add a reference to Fig. 1d to the end of L127. Is the mesh adjusted as the grounding line retreats?
- Note that the rainbow colour scheme in panel d is not in line with the journal guidelines.
- 39 The mesh is temporally static. We have clarified this in the manuscript and added a reference to
- 40 Fig. 2d. Additionally, we have updated the color scheme in Fig. 2d.
- zb, deep = Depth of the bedrock topography in Table 1
- 42 This is confusing, as your bedrock topography in Fig. 2 varies spatially. Consider: Maximum
- 43 depth of the initial bedrock topography.
- 44 Done.
- 45 L156: "these surface and basal forcings"
- 46 Since the lines directly above are discussing the melt rate, consider re-stating what these forcings
- are or refer to them as "perturbed surface and basal forcings."
- We have revised to "perturbed surface and basal forcings".

- 49 L169: "we choose the EAKF due to its reduced sensitivity to ensemble size"
- 50 This is somewhat confusing, as in L179, you state a common challenge with [...] EAKFs [...] arises
- 51 when the size of the ensemble is significantly smaller than the independently observed degrees of
- 52 freedom.
- We have added "compared to stochastic EnKFs" for clarity.
- 54 *L282: EnKF*
- 55 Should this be EAKF?
- 56 We have revised it to "EAKF".
- 57 L297: "continues to decrease steadily"
- 58 It does not continue to decrease steadily.
- 59 We have removed this.
- 60 L301-302: "the benefits saturate as the ensemble size increases from 50 to 100"
- Be more precise here. For bed elevation and ice thickness, an ensemble size of 100 has a larger
- RMSE at t = 30 a than for a size of 50.
- We have revised this sentence.
- 64 "While increasing the ensemble size from 30 to 50 shows clear improvements in DA performance,
- 65 further increasing the size to 100 does not consistently reduce RMSEs and, in some cases, even
- 66 results in slightly higher errors.."
- 67 L302: "illustrative purposes"
- 68 What do you mean by that?
- We chose this configuration as a representative setup to demonstrate performance.
- 70 "For the remaining twin experiments, we proceed with an ensemble size of 50, a localization radius
- 71 of 4 km, and an inflation value of 1.12 as a representative setup to demonstrate performance."
- Fig. 7: What causes the sharp grounding line extent towards higher x values at  $y=10\,\mathrm{km}$  in the
- 73 no assimilation panel?

- 74 You did not provide an answer to this question.
- 75 We revised the description of the model domain's initial condition to address this point:
- 76 "A sharp grounding line advance is observed near y = 10 km, likely caused by low surface elevation and high spatial variability in the underlying bed topography in this area."
- We chose to show the difference in ice thickness in Fig. 8 because changes in thickness are difficult to detect visually from the similar figure as Fig. 6 and 7.
- Yes, but wouldn't it also be easier to detect the changes if you plotted the differences in Figs. 6 and 7?
- We appreciate the reviewer's suggestion. While agree that plotting the difference in Figs. 6 and 7 could further aid interpretation, we chose to keep the original figure to preserve the spatial context of the results and allow direct comparison of the physical variables between experiments. Our intention in showing only the difference in ice thickness in Fig. 8 was to highlight subtle variations that may be difficult to detect visually in plots such as Figs. 6 and 7.
- 87 L362: "although RMSE values continue to decrease until the end of the assimilation window"
- 88 At 15 km across-track spacing, not all RMSE values continue to decrease.
- 89 We have removed this.
- We added the reference to Fig. 13 and described the increase in RMSE for the 20 km grid data.
- 91 Fig. is missing for some of these references.
- 92 We have revised those references.
- 93 L437: "observations years"
- 94 Change to "observation years."
- 95 Done.

- Estimation of the state and parameters in ice sheet model using an ensemble Kalman filter and Observing System 2
- Simulation Experiments 3
- Authors' response (RC2) -
- Youngmin CHOI et al.

5

6

11

16

17

18

21

22

23

- September 23, 2025
- I am satisfied with the substantial changes made by the authors throughout the manuscript in response to my first review. I am happy for the manuscript to proceed to publication if the authors are able to revisit one point.

In my original review, I suggested that if the authors were to claim that EAKF with DART is more computationally tractable than adjoint-based variational DA methods, then this needed to be backed up with computational benchmarking. The authors have since softened this claim and indicate that a direct computational comparison is beyond the scope of this study. I find this to be an acceptable response. However, it is still the case that if this study is meant to provide a proof-of-concept for the utility of using DART and ensemble-based methods more broadly, having 15 some information about the computational resources required for such methods would seem necessary so that readers who might think of using such an approach would know ahead of time what computational resources are required. Specifically, it would be useful if the authors could provide some information about the computational overhead associated with running DART for some of 19 these simulations. In principle, the assembly and computation of the analysis in Kalman-based DA methods can be done with relatively little computational expense compared to the forecast. However, in practice this requires gathering output from difference processes (if the ensemble members are run in parallel, which is what makes ensemble-based methods so promising from a computational perspective) and then calculating information on potentially large arrays. This can prove to be a challenge from an I/O and memory perspective as problems scale in size. I'm not asking for the authors to add a whole new section on this, but simple to use their existing ensemble simulations to estimate the marginal computational cost associated with the analysis step. Even just comparing the total walltime of two cases with the same number of ensemble members run on the same number of processes, but with more assimilation cycles would help to estimate the added

computational cost of each analysis cycle. This can be described in a few sentences added in the results or discussion section and shouldn't require running any new simulations assuming that the existing run logs already include information about total run time.

We thank the reviewer for this helpful suggestion. Based on our run logs, we compared simulations with identical ensemble sizes and model configuration but with different number of assimilation cycles. We have added a short description of these results to the discussion section.

"To provide a sense of computational resources for ensemble DA, we compared runs with identical 36 ensemble sizes but different numbers of assimilation cycles over the same 30-year assimilation 37 window: 30 annual cycles versus 60 semiannual cycles. For an ensemble size of 50 on a Broadwell 38 node with 28 cores, the 30-cycle run required approximately 2.9 hours of walltime, while the 60-39 cycle run required approximately 5.2 hours. Because both runs cover the same forecast period, the additional cost in the 60-cycle case reflects the more frequent execution of the DART analysis step 41 and associated I/O (e.g., conversion between ISSM and DART). A direct comparison of per-cycle 42 times is not strictly meaningful, since the forecast interval in each cycle is shorter in the semiannual 43 case, but the nearly doubled walltime for the 60-cycle case suggests that the analysis step and 44 associated I/O represent a significant fraction of the total computational time. As a result, scaling 45 to larger ensembles or higher-frequency updates will likely increase computational demands due to the analysis step and its I/O requirements."